# Down the Penrose stairs, or how selection for fewer recombination hotspots maintains their existence

Zachary Baker[1,2]*[†], Molly Przeworski[1,2,3], Guy Sella[2,3]

[1]Department of Systems Biology, Columbia University, New York, United States; [2]Department of Biological Sciences, Columbia University, New York, United States; [3]Program for Mathematical Genomics, Columbia University, New York, United States

*For correspondence:
zb267@cam.ac.uk

Present address: [†]Department of Genetics, Cambridge University, Cambridge, United Kingdom

Competing interest: The authors declare that no competing interests exist.

**Abstract** In many species, meiotic recombination events tend to occur in narrow intervals of the genome, known as hotspots. In humans and mice, double strand break (DSB) hotspot locations are determined by the DNA-binding specificity of the zinc finger array of the PRDM9 protein, which is rapidly evolving at residues in contact with DNA. Previous models explained this rapid evolution in terms of the need to restore PRDM9 binding sites lost to gene conversion over time, under the assumption that more PRDM9 binding always leads to more DSBs. This assumption, however, does not align with current evidence. Recent experimental work indicates that PRDM9 binding on both homologs facilitates DSB repair, and that the absence of sufficient symmetric binding disrupts meiosis. We therefore consider an alternative hypothesis: that rapid PRDM9 evolution is driven by the need to restore symmetric binding because of its role in coupling DSB formation and efficient repair. To this end, we model the evolution of PRDM9 from first principles: from its binding dynamics to the population genetic processes that govern the evolution of the zinc finger array and its binding sites. We show that the loss of a small number of strong binding sites leads to the use of a greater number of weaker ones, resulting in a sharp reduction in symmetric binding and favoring new PRDM9 alleles that restore the use of a smaller set of strong binding sites. This decrease, in turn, drives rapid PRDM9 evolutionary turnover. Our results therefore suggest that the advantage of new PRDM9 alleles is in *limiting* the number of binding sites used effectively, rather than in increasing net PRDM9 binding. By extension, our model suggests that the evolutionary advantage of hotspots may have been to increase the efficiency of DSB repair and/or homolog pairing.

## Editor's evaluation

This paper presents a theoretical model of the evolutionary dynamics of Prdm9-dependent meiotic recombination hotspots. This study provides important insights. It shows that selection acts to limit the number of hotspots and to increase hotspot symmetry. This is consistent with the proposed role of PRDM9 in coordinating DSB formation and repair. Although the authors did not explore all possible scenarios, the conclusions are convincing and open up directions for extending the model and testing some of its predictions.

## Introduction

Meiotic recombination is initiated by the deliberate infliction of double strand breaks (DSBs) across the genome and resolved with their subsequent repair using homologous chromosomes. In many species, including all plants, fungi and vertebrates investigated to date, the vast majority of meiotic recombination events localize to narrow intervals of the genome known as hotspots (reviewed in *Tock*

*and Henderson, 2018*). Why recombination is concentrated in hotspots in some taxa, as opposed to being more uniformly spread across the genome, as in *Drosophila* (*Manzano-Winkler et al., 2013*; *Smukowski Heil et al., 2015*) or *C. elegans* (*Kaur and Rockman, 2014*; *Bernstein and Rockman, 2016*), remains unknown. Moreover, among species with hotspots, there exists a great deal of variation in how hotspots are localized, and it is unclear know why these different mechanisms have been adopted in different lineages.

In humans and mice, and likely many other metazoan species, the primary determinant of hotspot locations is PRDM9 and the DNA-binding specificity of its Cys-2-His-2 Zinc Finger (ZF) array (*Myers et al., 2010*; *Baudat et al., 2010*; *Parvanov et al., 2010*; *Baker et al., 2017*). Early in meiosis, PRDM9 binds to thousands of binding sites along the genome (*Grey et al., 2017*; *Parvanov et al., 2017*), where its PR/SET domain catalyzes the formation of H3K4me3 and H3K36me3 marks on nearby histones (*Hayashi et al., 2005*; *Grey et al., 2011*; *Wu et al., 2013*; *Eram et al., 2014*; *Powers et al., 2016*). These marks, and protein interactions mediated by the PRDM9 KRAB domain, play a role in bringing PRDM9-bound sites to the chromosomal axis, where the SPO11 protein catalyzes the formation of DSBs at a small subset (*Imai et al., 2017*; *Parvanov et al., 2017*; *Diagouraga et al., 2018*; *Thibault-Sennett et al., 2018*; *Bhattacharyya et al., 2019*).

Whereas in species lacking PRDM9, such as canids, birds, fungi and plants, recombination rates are increased near promoter-like features, such as transcriptional start sites or CpG islands (*Auton et al., 2013*; *Singhal et al., 2015*; *Lam and Keeney, 2015*; *Yelina et al., 2015*) and the recombination landscape is conserved over tens of millions of years (*Singhal et al., 2015*; *Lam and Keeney, 2015*), in species that use PRDM9 to direct recombination, recombination rates are not elevated near such features and the recombination landscape is rapidly-evolving. Notably, the hotspots used in closely related species, such as humans and chimpanzees, or even different subspecies, show no more overlap than expected by chance (*Ptak et al., 2004*; *Winckler et al., 2005*; *Auton et al., 2012*; *Brick et al., 2012*; *Stevison et al., 2016*).

This rapid evolution of PRDM9 mediated recombination is driven by both the evolutionary erosion of PRDM9 binding sites due to gene conversion (GC) and concomitant turnover in the DNA-binding specificity of PRDM9's ZF-array. When individuals are heterozygous for 'hot' and 'cold' alleles, defined as those more or less likely to experience a DSB during meiosis, GC leads to the under-transmission of hot alleles in a manner that is mathematically analogous to purifying selection against them (*Nagylaki and Petes, 1982*; *Jeffreys and Neumann, 2002*). This phenomenon is observed in the over-transmission of polymorphisms disrupting PRDM9 binding motifs within hotspots (*Berg et al., 2011*; *Cole et al., 2014*). In the absence of any countervailing selective pressures, the under-transmission of hot alleles should lead to their rapid loss from the population over time (*Nicolas et al., 1989*; *Boulton et al., 1997*; *Pineda-Krch and Redfield, 2005*; *Coop and Myers, 2007*; *Peters, 2008*); this prediction too is met, in the unusually rapid erosion of PRDM9 binding sites in primate and rodent lineages (*Myers et al., 2010*; *Baker et al., 2015*; *Smagulova et al., 2016*; *Spence and Song, 2019*).

The use of hotspots persists despite their erosion, owing to evolutionary turnover in the DNA-binding specificity of PRDM9. Variation in hotspot usage has been associated with different PRDM9 alleles (i.e. with different amino acids at the DNA-binding residues of their ZFs) in mice, humans and cattle (*Myers et al., 2010*; *Baudat et al., 2010*; *Parvanov et al., 2010*; *Berg et al., 2010*; *Kong et al., 2010*; *Berg et al., 2011*; *Fledel-Alon et al., 2011*; *Hinch et al., 2011*; *Sandor et al., 2012*; *Ma et al., 2015*), and the experimental introduction of a PRDM9 allele with a novel ZF-array into mice reprograms the locations of hotspots (*Davies et al., 2016*). Moreover, strong signatures of positive selection on the DNA-binding residues of ZFs from PRDM9 genes suggests a rapid turnover of PRDM9 binding specificity driven by recurrent selection for novel binding specificities (*Oliver et al., 2009*; *Thomas et al., 2009*; *Myers et al., 2010*; *Buard et al., 2014*; *Baker et al., 2017*).

Several theoretical models have been developed in order to explain the co-evolution of PRDM9 and its binding sites (*Ubeda and Wilkins, 2011*; *Latrille et al., 2017*). In these models, GC acts to remove PRDM9 binding sites over time, leading to a proportional reduction in the amount of PRDM9 bound and consequently of DSBs or crossovers (COs). These models further assume that having too few DSBs or COs reduces fitness, potentially because at least one CO is required per chromosome for proper disjunction. In these models, therefore, the persistence of hotspots is mediated by a *Red Queen* dynamic, in which younger PRDM9 alleles are favored over older ones because their binding sites have experienced less erosion due to GC.

The argument that new PRDM9 alleles are favored because they restore PRDM9 binding seems at odds with the nature of hotspots, namely the use of only a small proportion of the genome for recombination. In the extreme, if selection simply favored PRDM9 alleles with a greater number of binding sites, an optimal PRDM9 allele would be one that can bind anywhere in the genome, at which point the heats of individual binding sites would be so diluted as to no longer be hotspots.

Moreover, multiple lines of evidence suggest that in fact, the loss of PRDM9 binding sites does not imperil the initiation of recombination events and only alters their locations. PRDM9-bound sites appear to compete for DSBs (*Diagouraga et al., 2018*) and the number of DSBs formed per meiosis is tightly regulated independently of PRDM9 (*Kauppi et al., 2013*; *Yamada et al., 2017*). Similar numbers of DSBs form even in PRDM9-/- mice (*Hayashi et al., 2005*; *Brick et al., 2012*; *Mihola et al., 2019*) and in a PRDM9-/- human female (*Narasimhan et al., 2016*); in both cases, the localization of DSB appear to 'default' to those found of species lacking PRDM9. Importantly, PRDM9 binding sites themselves are known to compete for available PRDM9 molecules, such that the loss of binding sites may only weakly affect the number bound (*Billings et al., 2013*; *Baker et al., 2014*). If the loss of PRDM9-mediated hotspots does not imperil the initiation of recombination then why are new PRDM9 alleles that restore them favored in evolution?

In addition to localizing DSBs, PRDM9 has recently been shown to play a role in DSB repair, mediated by its symmetric recruitment to the same loci on homologous chromosomes during meiosis (*Smagulova et al., 2016*; *Davies et al., 2016*; *Gregorova et al., 2018*; *Hinch et al., 2019*; *Li et al., 2019*; *Huang et al., 2020*; *Mahgoub et al., 2020*; *Wells et al., 2020*; *Gergelits et al., 2021*; *Davies et al., 2021*). This secondary function was first suggested by studies of hybrid sterility in mice, in which particular crosses result in sterile F1 males (reviewed in *Forejt et al., 2021*). In these sterile hybrids, the PRDM9 allele of each parental subspecies preferentially binds to the non-parental background, because the strongest PRDM9 binding sites corresponding to the alleles in each parental subspecies have been eroded. The resulting asymmetric binding leads to meiotic arrest in pachytene, associated with widespread delays in DSB repair, asynaptic chromosomes, and defects in homolog pairing (*Baker et al., 2015*; *Smagulova et al., 2016*; *Davies et al., 2016*). Restoring PRDM9 binding symmetry, either through the introduction of a novel PRDM9 allele with functioning binding sites in either parental subspecies (*Davies et al., 2016*; *Davies et al., 2021*) or by partially restoring the homozygosity of binding sites on the chromosomes most prone to asynapsis (*Gregorova et al., 2018*), is sufficient to rescue the fertility of hybrids. These results indicate that fertility depends on having a small number of DSBs at sites symmetrically bound by PRDM9 (*Gregorova et al., 2018*) on each chromosome.

The importance of DSBs at symmetrically bound sites may be related to them being repaired more rapidly—possibly because symmetric binding reduces the search space during homolog engagement and/or because chromatin modifications create a more accessible substrate for strand invasion—and the preferential use of sites that are repaired early for COs and homolog pairing. Genome-wide analyses indicate that DSBs at asymmetrically-bound hotspots are more likely to experience delays in their repair (*Hinch et al., 2019*; *Li et al., 2019*), that they are less likely to result in COs, possibly due to this delay (*Hinch et al., 2019*), and that they are more likely to be resolved using the sister chromatid rather than the homologous chromosome as a template (*Li et al., 2019*; *Gergelits et al., 2021*). Recent work has further implicated the gene *ZCWPW1* in mediating efficient repair at symmetrically bound sites (*Huang et al., 2020*; *Mahgoub et al., 2020*; *Wells et al., 2020*). These observations suggest that the selection favoring new PRDM9 alleles stems from the importance of symmetric binding in DSB repair, rather than DSB initiation, as previously envisioned.

In this view, the erosion of the strongest hotspots leads to a decline in symmetrically bound sites because of the competitive binding of PRDM9: the erosion of the few strong binding sites shifts binding toward a greater number of weaker ones, which are much less likely to be bound symmetrically. Based on this premise, we model the co-evolution of PRDM9 and its binding sites from first principles, assuming, as seems realistic, that: (i) the number of DSBs is regulated independently of PRDM9, (ii) PRDM9 binding is competitive between sites (i.e. the availability of PRDM9 is limiting), and (iii) fitness is a function of the probability that the smallest chromosome experiences some minimal number of DSBs at sites symmetrically bound by PRDM9.

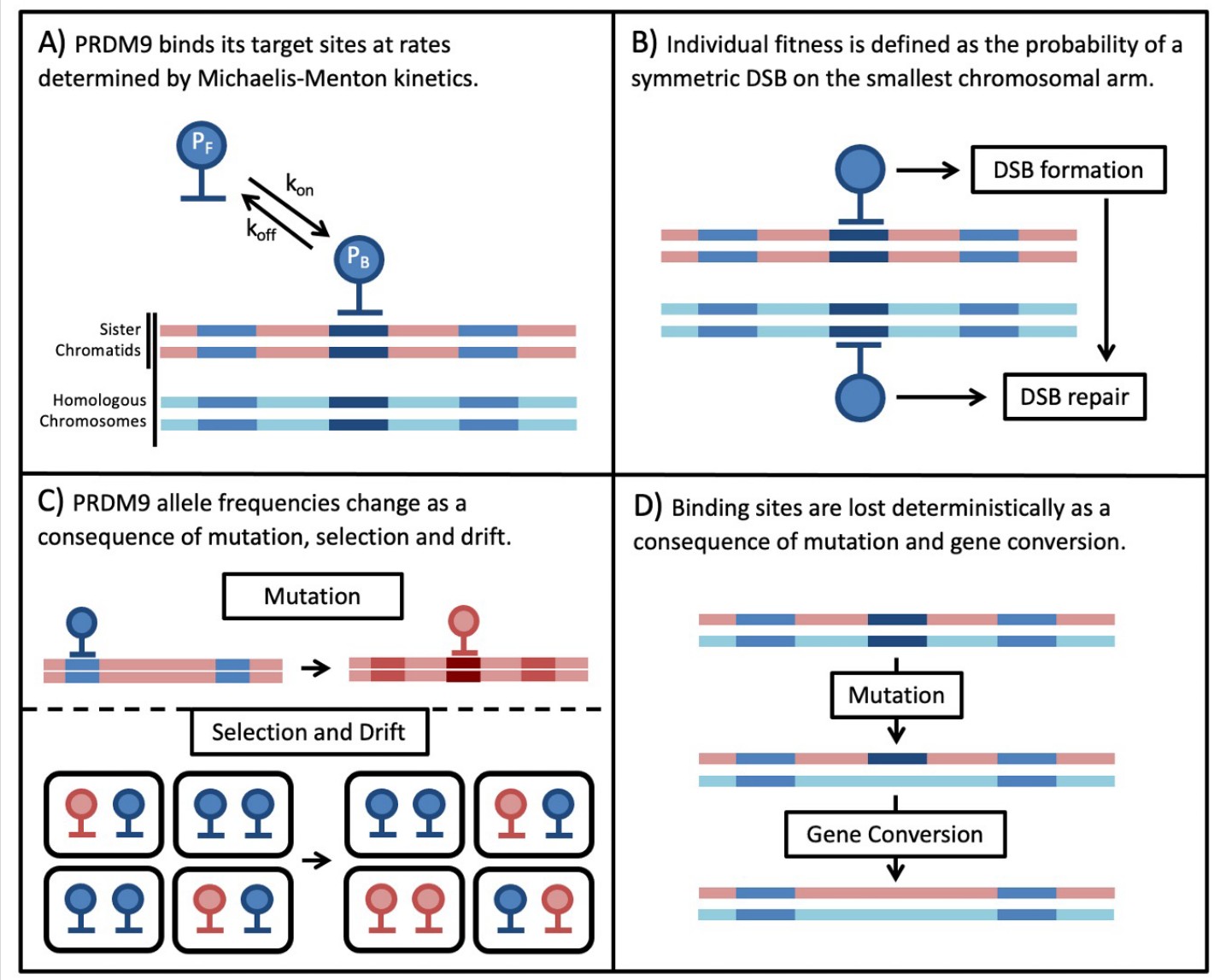

**Figure 1.** Overview of the Model.

## Model

### Overview

We model the co-evolution of PRDM9 and its binding sites in a diploid, panmictic population of constant size. Individuals are described in terms of their two PRDM9 alleles. Each PRDM9 allele recognizes a distinct set of binding sites, which may have distinct binding affinities or 'heats' (i.e. different probabilities of being bound). The model is composed of four main parts (*Figure 1*). The first describes PRDM9 binding during meiosis in terms of Michaelis-Menten kinetics, in which sites compete for binding their cognate PRDM9 proteins. We use this model to approximate the probability that any given site on the four chromatids (i.e. the four copies of a chromosome present during meiosis) is bound by PRDM9 (*Figure 1A*). We then approximate the probabilities of any given site experiencing a DSB, and of any given locus experiencing a DSB while being symmetrically bound by PRDM9, that is of a *symmetric DSB* (*Figure 1B*).

Next, we model the population genetic processes. We use Wright-Fisher sampling with viability selection in a panmictic diploid population of size $N$ to describe the change in PRDM9 allele frequencies in the population (*Figure 1C*). New PRDM9 alleles arise from mutation with probability $\nu$ per

individual meiosis and establish a new set of binding sites. Selection on a PRDM9 allele arises from its impact on fitness, and more specifically, the need for at least one symmetric DSB per chromosome per meiosis. For simplicity, we consider only the smallest chromosome because fitness defects arising from asymmetry in PRDM9 binding should affect the smallest chromosomes first (assuming, as seems plausible, that they have fewer sites at which a symmetric DSB might be made). Also for simplicity, we assume that individual fitness is equal to the probability of a symmetric DSB in a single meiosis. In reality, we expect fitness to depend on the proportion of an individual's gametes that have a sufficient number of symmetric DSBs, but this fitness should be a monotonically increasing function of the probability per meiosis, and therefore we expect the qualitative conclusions to be unchanged. Lastly, we use a deterministic approximation to describe the erosion of binding sites over time (*Figure 1D*). Mutations that destroy the ability of a site to bind to its cognate PRDM9 are introduced with probability μ per individual meiosis and can fix in the population due to biased gene conversion. The

**Table 1.** Parameters and variables of the model.

| Parameter | Description | Value |
|---|---|---|
| $N$ | Diploid population size | $10^3$–$10^6$ |
| μ | Mutation rate from hot to cold alleles at PRDM9 binding sites per generation | $1.25 \times 10^{-7}$ |
| $\nu$ | Mutation rate at the PRDM9 locus per generation | $10^{-5}$ |
| $D$ | Number of DSBs initiated per meiosis | 300 |
| $P_T$ | Total number of PRDM9 molecules expressed per meiosis | 5,000 |
| $k_i$ | Dissociation constant of PRDM9 binding sites with affinity $i$ | 5–50 |
| $B$ | Probability that a conversion tract spans the PRDM9 binding motif | 0.7 |
| $r$ | Proportion of the genome corresponding to the smallest chromosome | 1/40 |

| Variable | Description | |
|---|---|---|
| $n_i^j$ | Number of PRDM9 binding sites with affinity $i$, recognized by PRDM9 allele $j$, across the genome | |
| $H_i^{j,k}$ | Probability that a given site with binding affinity $i$, recognized by PRDM9 allele $j$, is bound by PRDM9 in an individual with PRDM9 alleles $j$ and $k$ (*Equation 1*) | |
| $P_B^{j,k}$ | Total number of bound cognate PRDM9 molecules in an individual with PRDM9 alleles $j$ and $k$ | |
| $P_F^{j,k}$ | Number of free cognate PRDM9 molecules in an individual with PRDM9 alleles $j$ and $k$ per meiosis (*Equation 2*) | |
| $c^{j,k}$ | Probability that any PRDM9-bound site experiences a DSB in an individual with PRDM9 alleles $j$ and $k$ (*Equation 3*) | |
| $\alpha_i^{j,k}$ | Probability of a PRDM9 binding site with affinity $i$, recognized by PRDM9 allele $j$, being symmetrically bound by PRDM9 and experiencing a DSB in an individual with PRDM9 alleles $j$ and $k$ (*Equation 4*) | |
| $W^{j,k}$ | The fitness of an individual with PRDM9 alleles $j$ and $k$ (*Equation 5* for homozygotes, *Equation 6* for heterozygotes) | |
| $W^j$ | The marginal fitness of PRDM9 allele $j$ across possible genotypes (*Equation 7*) | |
| $f^j$ | The allele frequency of PRDM9 allele $j$ (*Equation 8*) | |
| $g_i^{j,k}$ | Probability or rate of a PRDM9 binding site with affinity $i$, recognized by PRDM9 allele $j$, experiencing gene conversion spanning the PRDM9 binding motif, in an individual with PRDM9 alleles $j$ and $k$ (*Equation 9*) | |
| $g_i^j$ | Probability or rate of a PRDM9 binding site with affinity $i$, recognized by PRDM9 allele $j$, experiencing gene conversion spanning the PRDM9 binding motif, across the population (*Equation 10*) | |

fixation probability depends on rates at which the binding site experiences a DSB and of GC. Below, we describe each part of the model in turn; for a summary of the notation see **Table 1**.

## Molecular dynamics of PRDM9 binding

We model the binding of PRDM9 as a consequence of Michaelis-Menten kinetics, wherein binding sites compete for association with a fixed number of PRDM9 molecules. In Appendix 1, we solve these dynamics at equilibrium, that is when the rate at which PRDM9 dissociates from sites with any given binding affinity ('heat') is equal to the rate at which PRDM9 binds those sites. Specifically, we show that the *heat* of a site, that is the probability that a site with binding affinity $i$ is bound by PRDM9, can be expressed as

$$H_i = \frac{P_F}{P_F + k_i},$$

(1)

where $k_i$ ($\equiv k_{off}/k_{on}$) is the site's dissociation constant and $P_F$ is the number of free PRDM9 molecules. In turn, we show that the expected number of free PRDM9 molecules satisfies the equation

$$P_F = P_T - \sum_{i=1}^{m} \frac{P_F}{P_F + k_i} 4n_i,$$

(2)

where $P_T$ is the total number of PRDM9 molecules, $m$ is the number of binding affinities, and $n_i$ is the number of sites with affinity $i$, with $i = 1, \ldots, m$. When we consider models with one or two types of binding sites, this equation becomes a quadratic or cubic in $P_F$, respectively, which we solve analytically (see Appendix 1). In the general case, this equation becomes a polynomial of degree $m + 1$, which can be solved numerically.

These equations clarify that the competition among binding sites for PRDM9 molecules is mediated through the number of free PRDM9 molecules, $P_F$. **Equation 2** shows that when the number of binding sites of any given affinity increases, $P_F$ decreases; in turn, **Equation 1** shows that when $P_F$ decreases, so the probability of binding any given binding site ($H_i$). The competition among binding sites is strongest when the bulk of PRDM9 molecules are bound (i.e. $P_F \ll P_T$), because any increase in the number of bound sites (due to an increase in their number or affinity) causes a substantial decrease in $P_F$ and thus a decrease in the probability of binding any given binding site.

In turn, the total number of PRDM9 proteins of a given type depends on whether an individual is homozygous or heterozygous for the corresponding PRDM9 allele. We assume that a fixed number of PRDM9 molecules are produced per allele. Consequently, both $P_F$ and the probability of binding of any site ($H_i$) will always be higher in individuals homozygous for the cognate PRDM9 allele than in heterozygotes, and the ratio of heats for sites with different binding affinities will not be the same across genotypes.

The difference in the probability of binding the cognate PRDM9 in homozygotes and heterozygotes and the degree of competition among binding sites will be small when PRDM9 is not limiting (i.e., when $P_F \gg P_T - P_F$), but can be substantial when PRDM9 is limiting (i.e., when $P_T \gg P_F$). Empirical observations suggest that PRDM9 is limiting, with given binding sites bound at substantially greater rates in homozygote individuals than in heterozygotes or hemizygotes (**Flachs et al., 2012**; **Baker et al., 2015**); however, the degree to which is it limiting remains unknown and might also vary with, for instance, the age of PRDM9 alleles (see Discussion). In the main case we study below (the 'two-heat model'), we assume model parameter values for which the bulk of PRDM9 molecules are bound (i.e., $P_T \gg P_F$) and thus the competition among binding sites is strongest; this choice simplifies our analysis by reducing the number of parameters that affect the dynamics (see Appendix 3).

## Molecular dynamics of DSBs and symmetric DSBs

Next, we approximate the probability that a PRDM9-bound site experiences a DSB in a given meiosis. We assume that the total number of DSBs ($D$) is constant and that all PRDM9 bound sites are equally likely to experience a DSB. Thus, if the total number of PRDM9-bound sites in an individual with PRDM9 alleles $j$ and $k$ is $P_B^{j,k}$ then the probability that a given bound site experiences a DSB is

$$c = D/P_B^{j,k}$$

(3)

This approximation assumes that conditional on being bound, PRDM9 molecules arising from different alleles are equally likely to recruit the DSB machinery, but allows for some degree of dominance as a consequence of differences in the numbers of sites bound by each allelic variant genome-wide. However, in the main case we explore below (the 'two-heat model'), our assumption that the vast majority of PRDM9 molecules are bound for all alleles results in negligible degrees of dominance arising from differences in genome-wide levels of binding.

In Appendix 2, we show that the probability that a site with binding affinity $i$ experiences a symmetric DSB, that is that at least one of the chromatids is bound by PRDM9 and experiences a DSB and that at least one sister chromatid of each homolog is bound by PRDM9, is

$$\alpha_i = 1 - \left(1 - cH_i^2 \left(2 - H_i\right)\left(2 - cH_i\right)\right)^2 \tag{4}$$

For clarity, we omit indexes corresponding to the genotype at the PRDM9 locus, which affect $\alpha_i$ through $c$ (*Equation 3*) and $H_i$ through $P_T$ (*Equation 2*).

## Evolutionary dynamics of PRDM9 alleles

We model individual fitness as the probability per meiosis that at least one symmetric DSB occurs on the smallest chromosome, which comprises a proportion $r$ of the genome. We make the simplifying assumption that the proportion of binding sites of any affinity found on that chromosome is also $r$, and remains fixed over time; the number of binding sites with binding affinity $i$ is therefore $rn_i$. We also assume that the probability of symmetric DSBs at different binding sites are independent (i.e., ignore interference between them). Under these assumptions, we approximate the fitness of a homozygote for PRDM9 allele $i$ as 1 minus the probability that none of the binding sites on the smallest chromosome experience a symmetric DSB, that is

$$W^{j,j} = 1 - \prod_{i=1}^{m} \left(1 - \alpha_i^{j,j}\right)^{rn_{i,j}} \tag{5}$$

where the product is taken over the $m$ possible binding affinities and $n_{i,j}$ is the number of binding sites with affinity $i$ associated with PRDM9 allele $j$. Similarly, we approximate the fitness of an individual heterozygous for PRDM9, carrying alleles $j$ and $k \neq j$, as

$$W^{j,k} = 1 - \prod_{i=1}^{m} \left(1 - \alpha_i^{j,k}\right)^{rn_{i,j}} \left(1 - \alpha_i^{k,j}\right)^{rn_{i,k}} \tag{6}$$

The marginal fitness associated with a PRDM9 allele is calculated as a weighted average over genotypic frequencies, in which homozygotes for the focal allele, $j$, are counted twice to account for the two copies of the allele, that is

$$W^j = \sum_k f_k W^{j,k} \tag{7}$$

where $f_k$ is the frequency of PRDM9 allele $k$. The expected frequencies of allele $j$ after viability selection is then given by

$$f_j' = \frac{W^j}{\overline{W}} f_j \tag{8}$$

where $\overline{W}$ denotes the population's mean fitness. These expectations are used in the Wright-Fisher sampling across generations.

## Evolutionary dynamics of PRDM9 binding sites

We model the loss of PRDM9 binding sites due to biased gene conversion deterministically. Cold alleles at a binding site arise in the population at a rate of $2N$ per generation, where $N$ is the population size and is the mutation rate from a hot to a cold allele per gamete per generation. Biased gene conversion acts analogously to selection against hot alleles with a selection coefficient equal to the rate of gene conversion fully spanning that binding site, $g$ (*Nagylaki and Petes, 1982*). Consequently, a newly arisen cold allele (which is never bound by PRDM9) will eventually fix in the population with

probability $2g$. In turn, we model the probability $g_i^{j,k}$ that a binding site with affinity $i$, recognized by PRDM9 allele $j$, experiences gene conversion during a given meiosis in an individual with PRDM9 alleles $j$ and $k$ as the product of the probabilities that (i) it is bound by PRDM9 ($H_i^{j,k}$) (ii) it experiences a DSB conditional on being bound ($c^{j,k}$), and (iii) the resulting gene conversion spans the binding site ($B$), that is

$$g_i^{j,k} = Bc^{j,k}H_i^{j,k} \tag{9}$$

The expected population rate of gene conversion at this binding site is the average over individual rates, weighted by genotype frequencies (after viability selection), that is

$$g_i^j = 2f^j \sum_k f^k g_i^{j,k} - (f_j)^2 g_i^{j,j} \tag{10}$$

We model the fixations of cold alleles as if they occur instantaneously after they arise as mutations, with probabilities that derive from the rate of gene conversion at that time. Thus, we approximate the reduction in the number of binding sites with affinity $i$ for PRDM9 allele $j$ in a given generation by

$$\Delta n_{i,j} = 4N\mu g_i^j n_{i,j} \tag{11}$$

Because we assume that the number of DSBs and thus of gene conversion events per generation is constant, the total rate at which binding sites are lost, across all heats and PRDM9 alleles, is constant, and equals

$$\Delta n = \sum_j \sum_i \Delta n_{i,j} = N\mu BD \tag{12}$$

## Simulations

Simulations of the model were implemented using a custom R script (available at https://github.com/sellalab/PRDM9_model, copy archived at *Sellalab, 2023*). The simulation keeps track of the frequencies of PRDM9 alleles and of the numbers of binding sites of each considered heat for each allele. Each generation, the simulation calculates the marginal fitness of each PRDM9 allele (*Equation 7*) and the number of sites of each heat lost due to gene conversion (*Equation 11*). The frequencies of PRDM9 alleles for the succeeding generation are generated by Wright-Fisher (multinomial) sampling with viability selection (*Equation 8*). The number of new PRDM9 alleles generated by mutation each generation is sampled from a binomial distribution, where the probability of success is the mutation rate at PRDM9 $\nu$, and the number of trials is $2N$. These mutations are randomly chosen to replace pre-existing PRDM9 alleles (whose frequencies are correspondingly decreased by $1/2N$ per new mutation). In every generation, the program records various quantities discussed below, such as the population average fitness or heterozygosity at the PRDM9 locus.

## Estimated parameters

Except where stated otherwise, the parameters used in simulations are given in *Table 1*. The mutation rate from hot to cold alleles is taken to be 1.25x10$^{-7}$ per gamete per generation, to reflect a per base pair rate of 1.25x10$^{-8}$, as estimated in humans at typical reproductive age (*Kong et al., 2012*), and ~10 non-degenerate base pairs for PRDM9 binding per binding site, roughly consistent with observed binding motifs in mice and humans (*Baudat et al., 2010*; *Myers et al., 2010*). The mutation rate of new PRDM9 alleles is taken to be 10$^{-5}$ per gamete per generation, based on an estimate in humans (*Jeffreys et al., 2013*). The number of DSBs per meiosis is set to 300, in light of estimates in both humans and mice (*Baudat and de Massy, 2007*). In the main text, we set the number of PRDM9 molecules expressed during meiosis is set to 5,000, in line with an estimate of 4,700+/-400 in B6 mice, as inferred from comparing H3K4me3 ChIP-seq DNA at PRDM9-dependent hotspots with MNase treated input DNA (*Baker et al., 2014*), and roughly consistent with cytological observations of thousands of PRDM9 foci in the nuclei of meiotic cells from mice (*Parvanov et al., 2017*; *Grey et al., 2017*), under an assumption that most PRDM9 molecules are bound. Given that the estimate from *Baker et al., 2014* plausibly underestimates the amount of bound PRDM9, in Appendix 4, we repeat our main analyses where the number of PRDM9 molecules is set to 500, 1000 and 2500. We set the

probability that a gene conversion tract spans the PRDM9 binding motif to 0.7, based on estimates from mice (*Li et al., 2019*). Lastly, we set the proportion of the genome corresponding to the smallest chromosome to be 1/40, roughly equivalent to that of the smallest chromosome in mice.

## Results

### Fitness in models with one heat

Our model differs from previous ones (*Ubeda and Wilkins, 2011*; *Latrille et al., 2017*; *Úbeda et al., 2019*) in assuming that: (i) individual fitness is a function of the probability that a DSB is made on the smallest chromosome at a site symmetrically bound by PRDM9, (ii) PRDM9 binding sites compete for a finite number of PRDM9 molecules (i.e., that PRDM9 availability is limiting), and (iii) PRDM9-bound sites compete for a constant number of DSBs. In order to illustrate the importance of these assumptions, we first consider how fitness would change over time with or without these assumptions, under the simplest possible scenario, in which all individuals are homozygous for a single PRDM9 allele and all binding sites have the same binding affinity.

We assume that there are always more PRDM9-bound sites than DSBs per meiosis. If we drop assumptions (ii) and (iii), and instead posit that the number of bound PRDM9 molecules is proportional to the number of binding sites and that the number of DSBs is proportional to the number of bound PRDM9 molecules, fitness will decrease over time as binding sites are lost, as previous models predicted (*Figure 2A*: green line). This decrease in fitness is due to the dwindling numbers of DSBs per meiosis, and does not reflect changes in the probability of symmetric binding per site, which remains constant over time (*Figure 2B*: blue lines). If instead we make the more realistic assumption that PRDM9-bound sites compete for a fixed number of DSBs (*Diagouraga et al., 2018*), but continue to assume that PRDM9 binding is not competitive (i.e., dropping assumption (ii)), fitness does not change over time, because the number of DSBs per meiosis and the probability of symmetric binding per site remain constant (*Figure 2A and B*: blue lines). Importantly, in this case, there is no selection to drive the evolution of PRDM9. Lastly, we consider that both the availability of PRDM9 and DSBs are limiting, such that binding sites will compete PRDM9 molecules and PRDM9 bound sites will compete for DSBs (*Flachs et al., 2012*; *Baker et al., 2015*; *Diagouraga et al., 2018*). In this case, fitness increases over time (*Figure 2A*: red line), because, when binding sites are lost to BGC and their number decreases, the proportion of sites that are bound symmetrically increases (*Figure 2B*: red lines). This simple scenario illustrates in what way the predictions of our model diverge from those of previous ones.

Taken at face value, however, this toy analysis suggests that under our model, selection would favor and hence lead to the retention of older PRDM9 alleles, which is inconsistent with clear evidence for the rapid evolution of PRDM9 across taxa (*Baker et al., 2017*). As we discuss next, this prediction is a consequence of the unreasonable assumption that all binding sites have the same affinity.

### Fitness in a model with two heats

To understand how fitness might decay in more realistic settings, with competition among binding sites for PRDM9 molecules and among bound sites for DSBs, we turn to a model with two classes of binding sites. Specifically, we assume that there are a small number of strong binding sites that are bound symmetrically at substantial rates and a larger number of weaker binding sites, which are not. Consequently, recombination occurs almost entirely at the strong binding sites, whose numbers decrease over time and determine how fitness behaves; the weak binding sites, whose numbers do not change substantially over time, provide the backdrop of competitive PRDM9 binding. For the sake of clarity, henceforth we refer to the stronger class of binding sites as 'hotspots' and the weaker class as 'weak binding sites'.

The competitive effect of weak binding sites on individual hotspots is captured by the ratio of the dissociation constant at hotspots $k_1$ to the proportion of PRDM9 bound in the absence of any hotspots (see Appendix 3). We fix the number and dissociation constants of weak binding sites such that 99% of expressed PRDM9 molecules would bind the weak binding sites in the absence of hotspots, while contributing minimally to symmetric binding ($n_2 = 200,000$, $k_2 = 8030$). We then explore how the dynamics depend on the value of $k_1$.

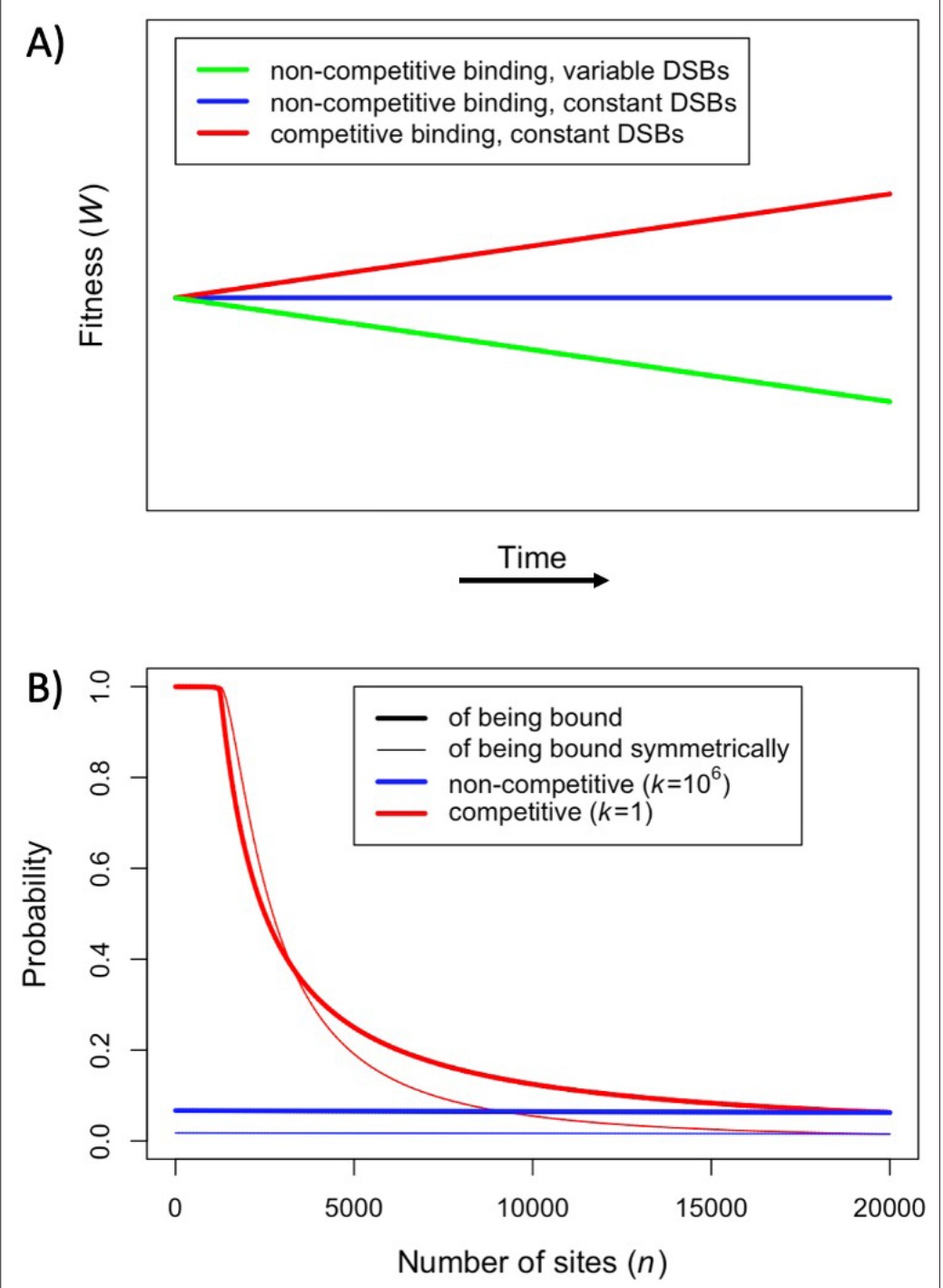

**Figure 2.** Fitness in the model with one heat. (**A**) Cartoon depicting the qualitative change in fitness as a function of time under different models, assuming that all individuals are homozygous for a single PRDM9 allele (without turnover) and that all binding sites have the same binding affinity. For visualization purposes, we present results for a linear function. (**B**) The probability of an individual binding site being bound (thicker lines) or of a given locus being bound symmetrically (thinner lines) if all sites are very weak and non-competitive ($k = 10^6$; blue) or very strong and competitive ($k = 1$; red). The values shown were computed using **Equation 1**, **Equation S2.2**.2 where, for sake of comparison, in both cases we set the number of PRDM9 molecules such that 5000 sites would be bound in the presence of 20,000 binding sites (roughly consistent with observations in mice **Parvanov et al., 2017**; **Grey et al., 2017**). When binding sites are very weak, most PRDM9 molecules are not bound and therefore our model behaves as if there were no competition between binding sites.

First, we consider how fitness depends on the number of hotspots, in individuals that are either homozygous or heterozygous for two PRDM9 alleles with the same binding distribution (*Figure 3*). (At this point, we do not yet consider the evolutionary dynamics, but gain insights helpful to elucidating these dynamics below.) Increasing the number of hotspots has conflicting effects on fitness: the proportion of bound PRDM9 localized to hotspots rises, and therefore there is a larger number of DSBs at hotspots, but the probability that any given hotspot is bound symmetrically decreases (*Figure 3—figure supplement 1*). Increasing PRDM9 expression has the opposite effects: it increases the probability that any given hotspot is bound symmetrically, but also increases the proportion of bound PRDM9, and thus DSBs, localized to weak binding sites (*Figure 3—figure supplement 1*). Consequently, for any value of $k_1$, there is an optimal number of hotspots that maximizes the probability of forming symmetric DSBs, which increases with PRDM9 expression levels (in order to counteract PRDM9 localization to weak binding sites). These considerations imply that the optimal number of hotspots per allele is smaller in heterozygotes relative to homozygotes (*Figure 3*). Similarly, the optimal number of hotspots is also smaller when hotspots are stronger, as stronger hotspots both increase the degree of symmetrical binding at hotspots, and result in less binding at weak binding sites (*Figure 3*).

When hotspots are relatively weak (e.g. $k_1 = 50$), fitness will always be higher in individuals homozygous for a given PRDM9 allele than in individuals heterozygous for two distinct PRDM9 alleles with the same number of hotspots (*Figure 3A*). Weak hotspots are far from being saturated for PRDM9 binding: the increased expression of a given PRDM9 allele in homozygotes relative to heterozygotes results in a notable increase in the probability that hotspots will be bound by PRDM9, while only slightly increasing the proportion of PRDM9 bound to weak binding sites (*Figure 3—figure supplement 1A*). Consequently, the benefit of the increase in symmetric binding at hotspots outweighs the cost of the slight increase in the proportion of DSBs localized to weak binding sites.

In contrast, when hotspots are relatively strong (e.g., $k_1 = 5$), fitness can be higher in individuals homozygous for a given PRDM9 allele or heterozygous for two similar alleles, depending on the number of hotspots recognized by the alleles (*Figure 3B*). When the number of hotspots far exceeds the number of PRDM9 molecules, hotspots are not saturated for PRDM9 binding, and fitness is always higher in homozygotes, as in the case with weaker hotspots. However, when there are fewer hotspots, those hotspots approach saturation for PRDM9 binding and consequently, the increased expression of a given PRDM9 allele in homozygotes relative to heterozygotes has a lesser impact on rates of symmetric binding (*Figure 3—figure supplement 1B*). Moreover, the cost associated with an increased proportion of PRDM9 localizing to weak binding sites becomes more pronounced (*Figure 3—figure supplement 1B*). The net effect is that, for PRDM9 alleles that recognize a sufficiently small number of hotspots, fitness is higher in heterozygotes than in homozygotes. Thus, in contrast to the case for weaker hotspots, individuals heterozygous for two PRDM9 alleles, each with the optimal number of hotspots for heterozygotes, will have higher fitness than individuals homozygous for a PRDM9 allele with the same number of hotspots (*Figure 3B*).

The relationship between the fitness of homozygotes and heterozygotes provides important insights into the evolutionary dynamics of our model. Consider a simplified model, with a population of infinite size, in which mutations to PRDM9 alleles with any number of hotspots arise every generation. If individuals heterozygous for a given mutant allele have greater fitness than any other individual in the population, that allele will quickly invade the population and rise to appreciable frequencies. Under these conditions, the PRDM9 alleles that successfully invade the population will necessarily be those with optimal numbers of hotspots in heterozygotes. When hotspots are sufficiently weak (e.g., $k_1 = 50$), the population exhibits cycles (*Figure 3A*): a PRDM9 allele with the optimal number of hotspots in heterozygotes invades the population and eventually fixes (arrow 1 in *Figure 3A*), after which individuals are homozygous for that allele and have greater fitness than any PRDM9 heterozygote, thereby preventing any other new allele from invading the population. Hotspots will then be lost from the population (arrow 2 in *Figure 3A*), until another PRDM9 allele can invade (arrow 3 in *Figure 3A*), fix and begin a new cycle. When hotspots are sufficiently strong (e.g., $k_1 = 5$), the population remains at a fixed point at which PRDM9 alleles with the optimal number of hotspots in heterozygotes are constantly invading (*Figure 3B*). Indeed, once an allele reaches appreciable frequency, individuals become homozygous for the allele; their fitness becomes lower than that of a heterozygote for the allele and a new PRDM9 allele with the optimal number of hotspots; and the

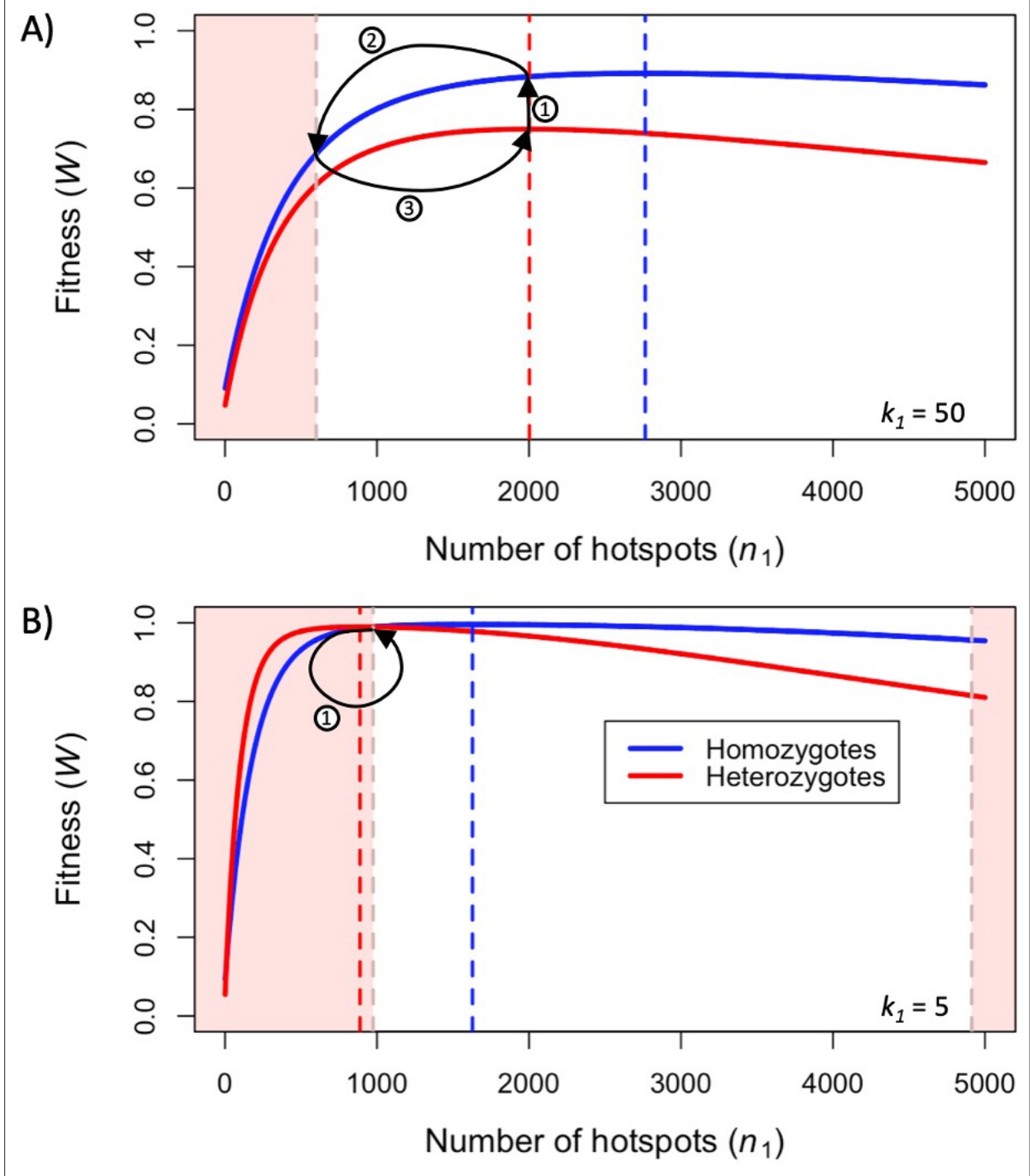

**Figure 3.** Fitness in the model with two heats. Fitness as a function of the number of hotspots, for an individual homozygous for a PRDM9 allele (blue), or heterozygous for two PRDM9 alleles with the same number of hotspots (red), when considering (**A**) weak hotspots or (**B**) strong ones. Fitness was calculated using *Equations 5 and 6*, assuming a backdrop of weak binding sites that would bind 99% of PRDM9 molecules in the absence of hotspots. Vertical dashed lines indicate the number of hotspots that maximize the fitness of individuals homozygous (blue) or heterozygous (red) for the allele. The area shaded in light red indicates the number of hotspots where the fitness of an individual homozygous for PRDM9 has lower fitness than the maximal fitness of a heterozygous individual that carries one copy of the same PRDM9 allele (where the other allele has the optimal number of hotspots for heterozygotes). In both panels, the region shaded in light red on the left reflects the case wherein the homozygous individual will have reduced fitness as a consequence of PRDM9 binding to weak binding sites. In panel B, the region on the right reflects the case wherein the homozygous individual will have reduced fitness as a consequence of limited symmetric binding at hotspots. The numbered arrows illustrate the behavior of the simplified evolutionary dynamic described in the text.

*Figure 3 continued on next page*

*Figure 3 continued*

The online version of this article includes the following figure supplement(s) for figure 3:

**Figure supplement 1.** The probability that a hotspot has been symmetrically bound (red), and the probability (or proportion) of DSBs localizing to hotspots (blue), when considering hotspots with (**A**) a relatively strong dissociation constant, or (**B**) a weaker one, under the two-heat model in individuals homozygous (solid lines) or heterozygous (dashed lines) for PRDM9.

allele is prevented from reaching fixation. These two kinds of behaviors, of cycles and a fixed point, are also seen under the more realistic evolutionary model that we consider next.

## Dynamics of the two-heat model

In particular, we consider the dynamic in a population of finite size, with a finite number of random mutations at the PRDM9 locus every generation. We assume that the number of hotspots associated with new PRDM9 alleles is uniformly distributed across a wide range ($n_1 = 1 - 5000$), which spans the optimal values for both homozygotes and heterozygotes across the range of hotspot dissociation constants explored ($k_1 = 5 - 50$). An alternative assumption would be to have the distribution of the initial number of hotspots tightly concentrated around the number of binding sites that best match the specificity of a typical PRDM9 ZF array; we explore the evolutionary dynamic of such a choice in **Appendix 5** and there we note the few places in which the assumption has a qualitative effect on our results.

Where possible, the values that we use for other model parameters are based on empirical observations in mice and humans, including the mutation rates at the PRDM9 locus and its binding sites, the number of DSBs initiated per meiosis, and the probability that a gene conversion event removes the hot allele at a PRDM9 binding site in individuals heterozygous for hot and cold alleles (**Table 1**). Given some uncertainty regarding estimates of the number of PRDM9 molecules expressed during meiosis, we explore the evolutionary dynamics with alternative values spanning a plausible range in **Appendix 4**; these changes do not have a qualitative effect on our results. Here, we use simulations (described in the model section) to study how the evolutionary dynamics depend on the dissociation constant at hotspots, $k_1$, which we know less about, and the effective population size, which varies over orders of magnitude in natural populations (**Leffler et al., 2012**).

When the population size is sufficiently large (e.g., $N = 10^6$), the dynamics are similar to that of the infinite population size case considered in the previous section (**Figure 4A and C**). When hotspots are fairly weak (e.g., $k_1 = 50$), they follow an approximate cycle (**Figure 4A**): a new PRDM9 allele that establishes a near optimal number of hotspots for heterozygotes invades and fixes, dominating the population in a homozygous state until enough hotspots are lost that the population becomes susceptible to invasion of a new allele that likewise establishes a near optimal number of hotspots for heterozygotes. Diversity at the PRDM9 locus is low throughout most of this cycle, with bouts of diversity in the period during which the dominant allele is taken over by a new one. Fitness peaks when a new allele is fixed and then decreases over time, as that allele loses its hotspots.

In contrast, when hotspots are strong (e.g., $k_1 = 5$), dynamics are approximately at a fixed point (**Figure 4C**): new PRDM9 alleles that lead to near optimal numbers of hotspots for heterozygotes continually invade but never fix. The fitness of individuals homozygous for such PRDM9 alleles is lower than that of individuals heterozygous for one such allele and those already present. Consequently, new PRDM9 alleles experience frequency-dependent selection, in which they are favored at low frequencies because they have near optimal numbers of hotspots for heterozygotes, but selected against at higher frequencies because of the reduced fitness of homozygotes. Diversity remains high and average fitness is approximately constant.

In smaller populations (e.g., $N = 10^3$), the dynamic is qualitatively similar to that in larger populations when hotspots are fairly weak (e.g., $k_1 = 50$; compare **Figure 4A and B**). The reduced mutational input at hotspots and correspondingly lower rate at which they are lost leads to longer intervals of time between fixation events. In turn, the reduced mutational input at PRDM9 and stronger drift leads to reduced peaks of PRDM9 diversity during those events.

In contrast, when hotspots are strong (e.g., $k_1 = 5$), dynamics differ markedly in smaller population compared to larger ones (compare **Figure 4C and D**): whereas in large populations, PRDM9 alleles never fix, in small populations, the lower mutational input at the PRDM9 locus and stronger genetic drift allow PRDM9 alleles to fix intermittently (**Figure 4D**). PRDM9 alleles with a small number of

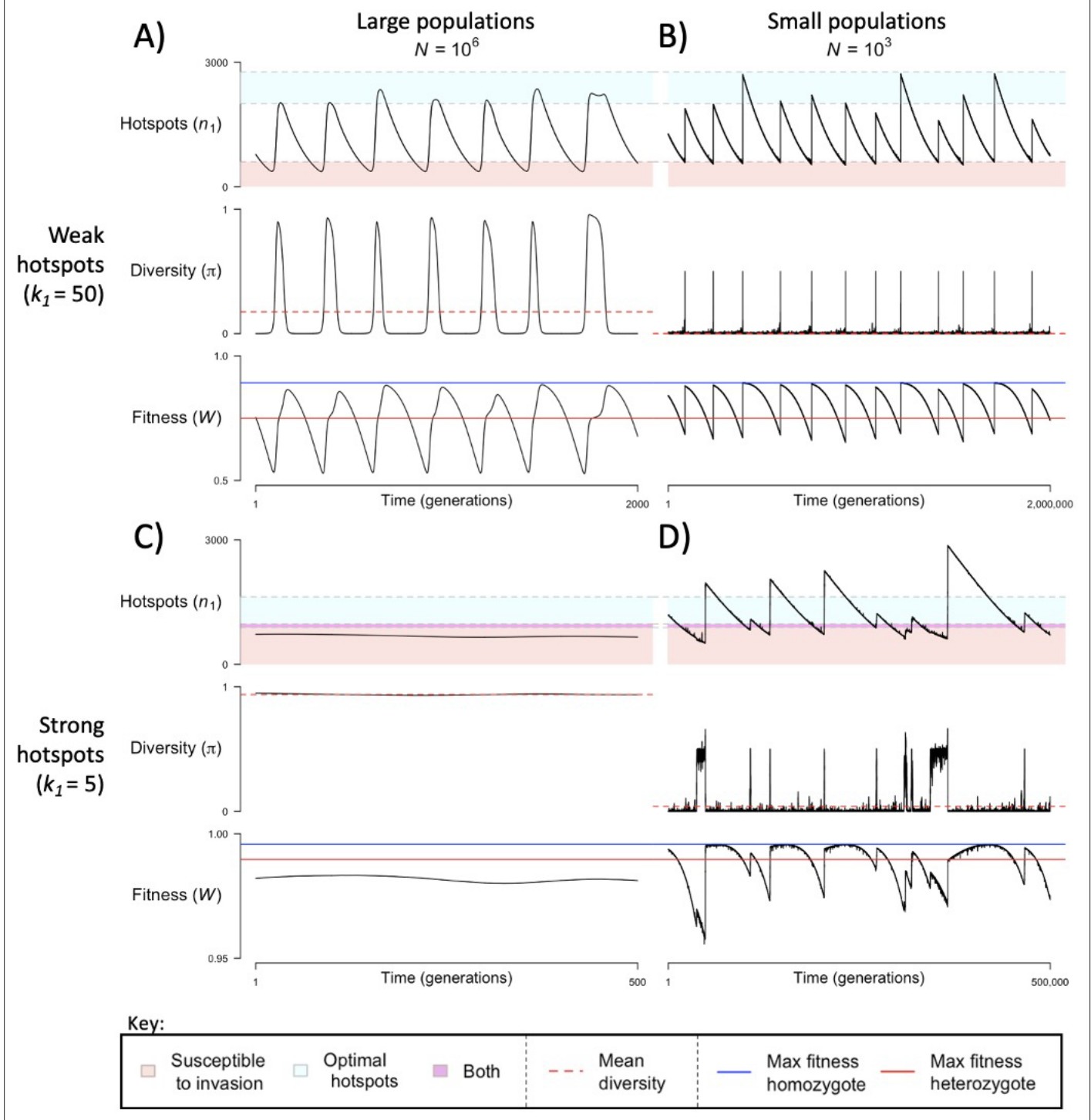

**Figure 4.** The dynamics of the model with two heats. The mean number of hotspots ($n_1$), diversity at PRDM9 ($\pi$), and mean fitness ($W$), as a function of time. We show four cases: with weak and strong hotspots (top and bottom rows, respectively) and large and small population size (left and right column, respectively). Simulations (as detailed in the model section) were run with the specified population sizes and hotspot dissociation constants, with other parameter values detailed in *Table 1*. The time (x-axis) in each plot has been scaled by the duration of PRDM9 turnover to allow for ~10 'cycles' (see *Figure 3*), which span ~$10^3$ times more generations in small populations compared to large ones. The highlighted range of hotspot numbers in which a PRDM9 allele is 'susceptible to invasion' corresponds to the regions shaded in red in *Figure 3*. In turn, the highlighted range of hotspot numbers designated as 'optimal hotspots' corresponds to the range between the number that optimizes fitness in heterozygotes and the number that optimizes fitness in homozygotes (between the dashed lines shown in *Figure 3*). See Appendix 4 for analogous results assuming an alternative value for the

*Figure 4 continued on next page*

*Figure 4 continued*

number of PRDM9 molecules expressed during meiosis ($P_T = 1000$), and Appendix 5 for analogous results assuming a tight unimodal distribution of the number of hotspots associated with newly arising PRDM9 alleles. This figure was generated using *Figure 4—source data 1–4*.

The online version of this article includes the following source data for figure 4:

**Source data 1.** The number of hotspots ($n_1$), average population fitness ($W$), and diversity at the PRDM9 locus (pi) per generation, across 55,000 simulated generations of the two-heat model with a large population size ($N = 10^6$) and weak hotspots ($k_1 = 50$).

**Source data 2.** The number of hotspots ($n_1$), average population fitness ($W$), and diversity at the PRDM9 locus ($\pi$) per generation, across 2,000,000 simulated generations of the two-heat model with a small population size ($N = 10^3$) and weak hotspots ($k_1 = 50$).

**Source data 3.** The number of hotspots ($n_1$), average population fitness ($W$), and diversity at the PRDM9 locus ($\pi$) per generation, across 55,000 simulated generations of the two-heat model with a large population size ($N = 10^6$) and strong hotspots ($k_1 = 5$).

**Source data 4.** The number of hotspots ($n_1$), average population fitness ($W$), and diversity at the PRDM9 locus ($\pi$) per generation, across 500,000 simulated generations of the two-heat model with a small population size ($N = 10^3$) and strong hotspots ($k_1 = 5$).

hotspots are typically lost rapidly and without reaching fixation, the same behavior as in larger populations. But when a mutation occurs to a PRDM9 allele with a large number of hotspots, such that homozygotes for the new allele have higher fitness than any heterozygotes (see *Figure 3B*), and the allele happens to reach high frequency by chance, it can be maintained in the population at appreciable frequency. Eventually, however, it loses a sufficient number of hotspots for the population to become susceptible to the invasion of a new PRDM9 allele. These chance fixation events and the corresponding number of hotspots leads to intermittent, chaotic cycles.

Regardless of the population size, stochasticity in the PRDM9 mutations that arise and escape immediate loss gives rise to a distribution of initial number of hotspots associated with segregating PRDM9 alleles. If the mutational distribution of the number of hotspots is tightly centered around a value dictated by the properties of the PRDM9 ZF array, then this distribution dictates the distribution of the initial number of hotspots for segregating alleles (see Appendix 5). If the mutational distribution is wider, as we assume here, then the distribution for segregating alleles is largely shaped by the evolutionary dynamics. We estimate this distribution in simulations by weighting the initial number of hotspots of PRDM9 alleles by the product of the alleles' sojourn times and mean frequencies; this weighting is equivalent to the probability of randomly sampling segregating alleles from the population (*Figure 5*). The distribution primarily reflects the number of hotspots of successfully invading alleles, and its mean is therefore near the optimal number of hotspots for heterozygotes (*Figures 5 and 6A*), although generally greater. Moreover, it is closer to the optimum in large populations than in small ones, because with greater mutational input at PRDM9, the alleles that invade in larger populations tend to be closer to the optimum (*Figure 6A*). By the same token, when hotspots are strong and the population size is small, the intermittent fixations of PRDM9 alleles with a large number of hotspots shift the distribution towards larger initial numbers of hotspots (*Figures 5 and 6A*).

The distribution of the number of hotspots associated with PRDM9 alleles by the time they exit the population also reflects the invasion dynamics of new PRDM9 alleles. We estimated this distribution in the same way that we did for the initial number (*Figures 5 and 6A*). Given that exiting PRDM9 alleles continue to lose hotspots during the time it takes for a new invading PRDM9 allele to arise and fix, the mean of the distribution is below the number at which the population becomes susceptible to invasion by heterozygotes carrying a new allele with the optimal numbers of hotspots. More hotspots are lost during this time in larger populations than in smaller ones, owing to the larger mutational input at binding sites and thus the greater rate of loss in larger populations. The turnover time of PRDM9 alleles is largely determined by the rate at which hotspots are lost and the number that are lost before a new allele becomes favored. We estimate the mean turnover time in simulations using the same weighting across alleles as described above. Turnover time is inversely proportional to population size (*Figure 6B*), because the increase in mutational input at binding sites and thus the rate at which they are lost are proportional to the population size (see *Equation 11*); other factors, such as the mutational input at PRDM9, have negligible effects by comparison (*Figure 6—figure supplement 1A*, *Figure 6—figure supplement 2*). The proportionality constant relating turnover time and population size depends on the heat of hotspots. The turnover time is shorter for strong hotspots than for weak ones (*Figure 6B*), because PRDM9 alleles with stronger hotspots enter the population with fewer hotspots (*Figure 6A*) and stronger hotspots experience more DSBs and are therefore lost faster.

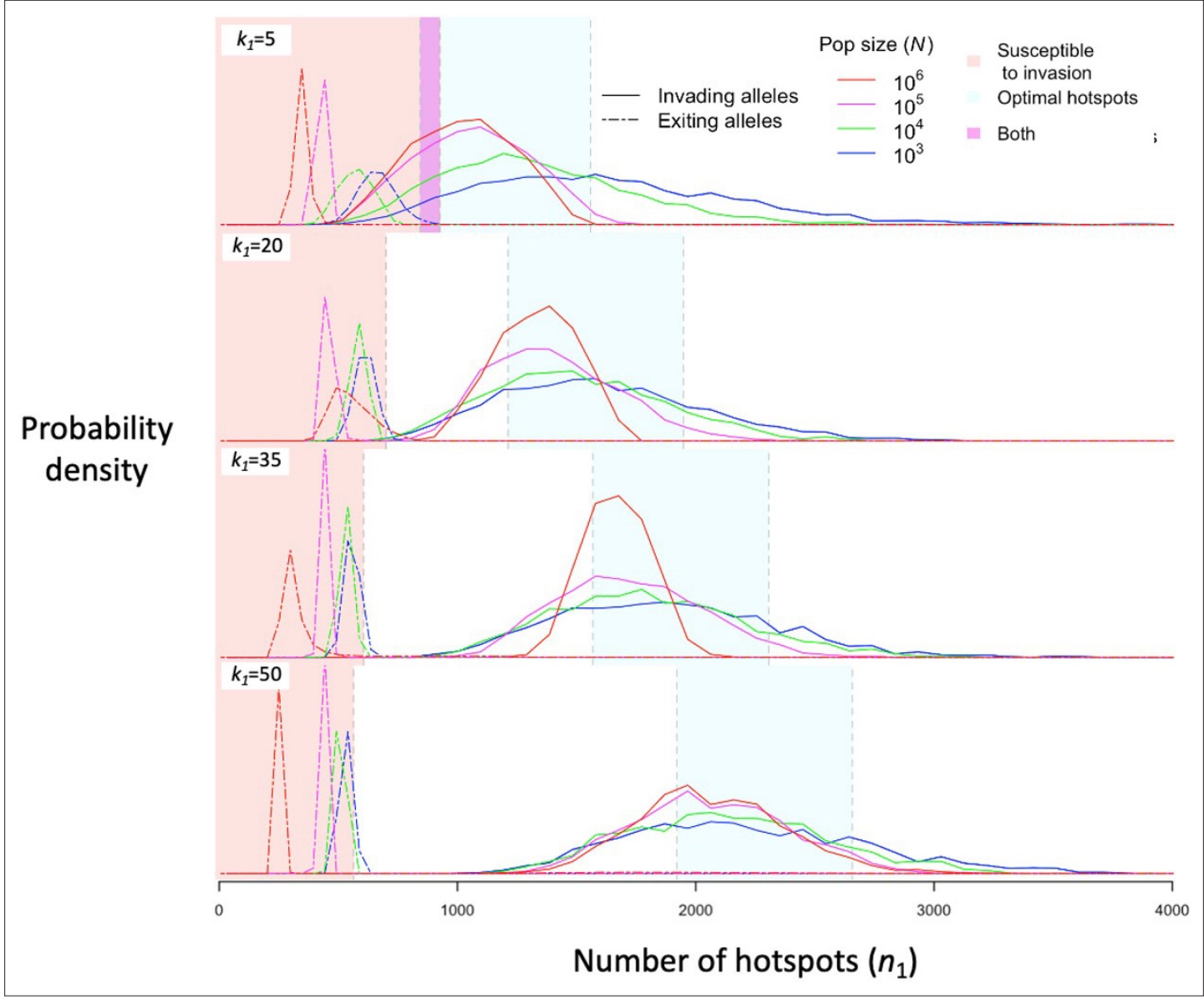

**Figure 5.** The distribution of initial and final numbers of hotspots among segregating PRDM9 alleles. The relative densities of initial and final numbers of hotspots (corresponding to invading and exiting alleles, respectively) for different dissociation constants (rows) and population sizes (line colors) obtained from simulations of five million PRDM9 allele trajectories with each set of parameters (see the model section for details about the simulations and *Table 1* for other parameter values). Shaded regions are defined the same way as in *Figure 4*. This figure was generated using *Figure 5—source data 1–2*.

The online version of this article includes the following source data for figure 5:

**Source data 1.** The distribution of the initial number of hotspots ($n_1$) among PRDM9 alleles, weighted by their mean allele frequency and sojourn time.

**Source data 2.** The distribution of the number of hotspots ($n_1$) among exiting PRDM9 alleles, weighted by their mean allele frequency and sojourn time.

The population size and dissociation constant affect diversity levels at the PRDM9 locus in several ways (*Figures 4 and 6C*). As shown in *Figure 4*, when hotspots are weak, PRDM9 evolution exhibits cycles, and PRDM9 diversity peaks at the phase in which new PRDM9 alleles invade and ascend to fixation. Average RDM9 diversity in this case can be approximated as the proportion of the cycle that makes up this invasion phase multiplied by the average diversity during this phase. The proportional length of the invasion phase increases when the turnover time is shorter, and the turnover time is shorter in larger populations (*Figure 6B*). Diversity during the invasion phase also increases with population size, owing to the greater mutational input at the PRDM9 locus (compare, e.g., *Figure 4A and*

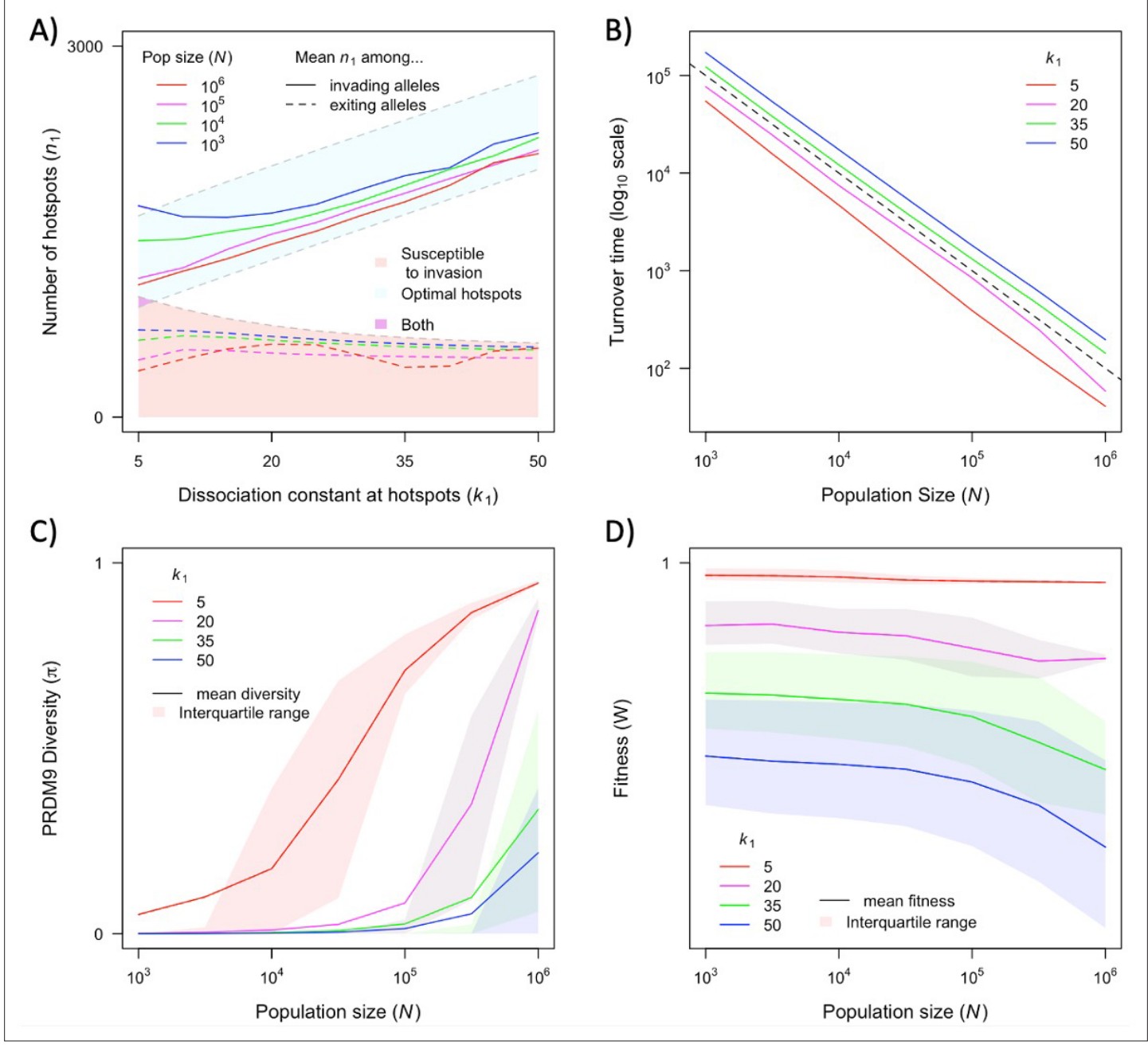

**Figure 6.** The dependence of key quantities on population size and hotspot heat. (**A**) Mean number of hotspots among incoming and exiting alleles (corresponding to the distributions shown in **Figure 5**). (**B**) Average turnover time of segregating PRDM9 alleles (see text). (**C**) Mean and interquartile range of diversity at PRDM9 over time. (**D**) Mean and interquartile range of fitness in the population. Each point in these graphs is based on simulations of $5 \times 10^6$ PRDM9 alleles (see model section for details) with the specified parameter values and the other model parameter values specified in **Table 1**. See Appendix 4 for analogous results, assuming alternative levels of PRDM9 expression. This figure was generated using **Figure 6—source data 1 and 2**.

The online version of this article includes the following source data and figure supplement(s) for figure 6:

**Source data 1.** Summary statistics from simulations for a range of population sizes ($N$) and hotspot dissociation constants ($k_1$), including mean turnover times, mean initial numbers of hotspots, and mean number of hotspots for exiting PRDM9 alleles (each weighted by the sojourn time of PRDM9 alleles).

**Source data 2.** Additional summary statistics from simulations for a range of population sizes ($N$) and hotspot dissociation constants ($k_1$), including mean diversity at PRDM9 ($\pi$) and population fitness ($W$), and the inter-quartile ranges of each.

**Figure supplement 1.** The effect of population size on the rate of turnover (**A**) and diversity (**B**) at PRDM9 while keeping population-scaled mutation rates at PRDM9, binding sites, or both, constant (when $k_1 = 20$).

*Figure 6 continued on next page*

*Figure 6 continued*

**Figure supplement 1—source data 1.** Mean PRDM9 turnover times for a range of population sizes ($N = 10^3$ to $10^6$) when considering moderately strong hotspots ($k_1 = 20$), under normal parameters, when the population-scaled mutation rate at PRDM9 has been normalized to that of a moderate population size ($N = 10^{4.5}$), when the population-scaled mutation rate at PRDM9 binding sites has been normalized to that of a moderate population size ($N = 10^{4.5}$), and when both mutation rates have been so normalized.

**Figure supplement 1—source data 2.** Mean values of diversity at the PRDM9 locus ($\pi$) for a range of population sizes ($N = 10^3$ to $10^6$) when considering moderately strong hotspots ($k_1 = 20$), under normal parameters, when the population-scaled mutation rate at PRDM9 has been normalized to that of a moderate population size ($N = 10^{4.5}$), when the population-scaled mutation rate at PRDM9 binding sites has been normalized to that of a moderate population size ($N = 10^{4.5}$), and when both mutation rates have been so normalized.

**Figure supplement 2.** The effect of population-scaled mutation rates on the dynamics of the model when hotspots are cold ($k_1 = 50$) or hot ($k_1 = 5$).

**Figure supplement 2—source data 1.** The number of hotspots ($n_1$), average population fitness ($W$), and diversity at the PRDM9 locus ($\pi$) per generation, across 2,000,000 simulated generations of the two-heat model with a small population size ($N = 10^3$) and weak hotspots ($k_1 = 50$), when the population-scaled mutation rate at PRDM9 was adjusted to match that of a large population (with $N = 10^6$; $\nu = 10^{-2}$).

**Figure supplement 2—source data 2.** The number of hotspots ($n_1$), average population fitness ($W$), and diversity at the PRDM9 locus ($\pi$) per generation, across 500,000 simulated generations of the two-heat model with a small population size ($N = 10^3$) and strong hotspots ($k_1 = 5$), when the population-scaled mutation rate at PRDM9 was adjusted to match that of a large population (with $N = 10^6$; $\nu = 10^{-2}$).

**Figure supplement 2—source data 3.** The number of hotspots ($n_1$), average population fitness ($W$), and diversity at the PRDM9 locus ($\pi$) per generation, across 2,000,000 simulated generations of the two-heat model with a small population size ($N = 10^3$) and weak hotspots ($k_1 = 50$), when the population-scaled mutation rate at PRDM9 binding sites was adjusted to match that of a large population (with $N = 10^6$; $\mu = 1.25 * 10^{-4}$).

**Figure supplement 2—source data 4.** The number of hotspots ($n_1$), average population fitness ($W$), and diversity at the PRDM9 locus ($\pi$) per generation, across 500,000 simulated generations of the two-heat model with a small population size ($N = 10^3$) and strong hotspots ($k_1 = 5$), when the population-scaled mutation rate at PRDM9 binding sites was adjusted to match that of a large population (with $N = 10^6$; $\mu = 1.25 * 10^{-4}$).

**Figure supplement 2—source data 5.** The number of hotspots ($n_1$), average population fitness ($W$), and diversity at the PRDM9 locus ($\pi$) per generation, across 55,000 simulated generations of the two-heat model with a small population size ($N = 10^3$) and weak hotspots ($k_1 = 50$), when the population-scaled mutation rate at both PRDM9 and PRDM9 binding sites was adjusted to match that of a large population (with $N = 10^6$; $\nu = 10^{-2}$; $\mu = 1.25 * 10^{-4}$).

**Figure supplement 2—source data 6.** The number of hotspots ($n_1$), average population fitness ($W$), and diversity at the PRDM9 locus ($\pi$) per generation, across 55,000 simulated generations of the two-heat model with a small population size ($N = 10^3$) and strong hotspots ($k_1 = 5$), when the population-scaled mutation rate at both PRDM9 and PRDM9 binding sites was adjusted to match that of a large population (with $N = 10^6$; $\nu = 10^{-2}$; $\mu = 1.25 * 10^{-4}$).

*B*). Consequently, average PRDM9 diversity is expected to be greater in larger populations. Moreover, in smaller populations, average PRDM9 diversity falls outside the interquartile range (*Figure 6C*), because during most of the cycle, PRDM9 diversity (outside the invasion phase) is very low. When hotspots are strong, there is a population size below which PRDM9 evolution is at steady state with continuously high PRDM9 diversity (with none of the alleles fixing) and above which the dynamics exhibit intermittent fixations and chaotic cycles (compare, e.g., *Figure 4C and D*). At lower population sizes with intermittent cycles, average PRDM9 diversity increases with population size and falls outside the interquartile range for the same reasons it does with weaker hotspots (*Figure 6C*). At higher population sizes with continuously high PRDM9 diversity, diversity levels increase with population size owing to the greater mutational input at the PRDM9 locus.

The cycling dynamics characteristic of PRDM9 evolution over much of the parameter space entails a substantial genetic load, i.e., an average fitness that is substantially lower than its optimal value (*Figure 6D*). This is a straightforward consequence of the number of hotspots associated with the predominant PRMD9 allele being substantially lower than the optimal number during most of the cycle (*Figures 5 and 6A*). Moreover, in contrast to the usual case, in this setting, average fitness is greater in smaller populations than in larger ones, as the number of hotspots – and thus the fitness associated with the predominant PRDM9 alleles when they fix and when they exit the population – is lower in larger populations than in smaller ones (*Figures 5 and 6A*).

## Discussion

Previous models of PRDM9 evolution demonstrated how selection to maintain a given rate of recombination (e.g., for there to be at least one crossover per chromosome) can act as a countervailing force to the loss of hotspots through gene conversion, and ensure the persistence of hotspots (*Ubeda*

*and Wilkins, 2011*; *Latrille et al., 2017*). These models were based on an assumption now believed to be incorrect, however: that the loss of PRDM9 binding sites leads to a reduction in the number of DSBs. Since DSB formation is regulated independently of PRDM9 (*Kauppi et al., 2013*; *Yamada et al., 2017*; *Brick et al., 2012*; *Mihola et al., 2019*) and hotspots are thought to compete for binding (*Diagouraga et al., 2018*), it more plausible that when hotspots are lost, PRDM9 simply binds elsewhere. Moreover, since development of these models, new lines of evidence indicate that the successful completion of meiosis requires the formation of DSBs at sites symmetrically bound by PRDM9 across homologous chromosomes, likely in order to assure accurate and efficient homolog pairing (*Smagulova et al., 2016*; *Davies et al., 2016*; *Gregorova et al., 2018*; *Hinch et al., 2019*; *Li et al., 2019*; *Huang et al., 2020*; *Mahgoub et al., 2020*; *Wells et al., 2020*; *Gergelits et al., 2021*; *Davies et al., 2021*).

Motivated by these discoveries, we developed a new model for the evolution of PRDM9 that accounts for the competition of binding sites for PRDM9 molecules and of PRDM9-bound sites for DSBs, in which the selection pressure stems from the requirement for having at least one DSB at a symmetrically bound site on the smallest chromosome. In our model, the selection driving the evolution of the PRDM9 ZF arises from the requirement of symmetric binding in DSB repair, and thus from changes in the shape of its binding distribution over time, rather than from changes in net binding, as previously assumed. In this setting, the loss of individual binding sites leads to increased PRDM9 binding and DSB formation at remaining sites. The rates at which binding sites are lost from the population due to gene conversion and the probabilities that they experience symmetric DSBs are each determined by their underlying affinities for PRDM9. Strong hotspots are lost more rapidly than weak ones, resulting in a shift from a small number of strong binding sites to a larger number of weak binding sites over the lifespan of a PRDM9 allele. This shift entails a reduction in the number of symmetric DSBs, and eventually selection in favor of a new PRDM9 allele. New PRDM9 alleles restore binding sites, thereby serving as a countervailing force to gene conversion. They are favored because, in doing so, PRDM9 binding is redirected to a smaller proportion of the genome, and is therefore more likely to bind symmetrically.

## Scope of the model

Our two-heat model allowed us to characterize a variety of qualitative behaviors and their dependence on evolutionary parameters, many of which are likely to hold for more complex and realistic distributions of heats. Yet the properties of this distribution will affect the quantitative predictions of the model, such as those about the relative fitness in homozygotes and heterozygotes for different PRDM9 alleles, which in turn shape the evolutionary dynamics. At present, we do not know enough about the distribution of heats (notably for newly arising PRDM9 alleles) in order to move toward reliable quantitative predictions. Existing inferences about this distribution do not estimate how much PRDM9 is bound at binding sites that fall below an arbitrary heat threshold (i.e., at sites which are not efficiently bound, but whose heat could increase after a sufficient number of stronger sites are lost) (*Baker et al., 2014*; *Grey et al., 2017*; *Altemose et al., 2017*). Moreover, given our current knowledge, the range of possibilities is too large for an exhaustive theoretical exploration to be feasible. What is needed are estimates of the distribution of heats for given PRDM9 alleles across different genetic backgrounds with varying degrees of erosion.

Also relevant is the number of PRDM9 molecules present in the relevant stage of meiosis and the fraction that are bound. In the two-heat model, we made the simplifying assumption that the vast majority of PRDM9 molecules are bound even in the absence of hotspots. This assumption may be too extreme. In B6 mice heterozygous for B6 and CST derived PRDM9 alleles, in which both alleles have similar mRNA expression levels, the vast majority of both PRDM9-dependent H3K4me3 marks and DSBs (~75%) localize to sites bound by the CST allele (*Flachs et al., 2012*; *Smagulova et al., 2016*; *Diagouraga et al., 2018*). This observation suggests that only one-third of available B6-derived PRDM9 molecules are actually bound. Nonetheless, PRDM9 binding is increased in B6 homozygotes relative to hemizygotes, indicating that PRDM9 binding is limiting in this case (*Baker et al., 2015*). In this regard, we note that recent pre-print by *Genestier et al., 2023* explores a model that is similar to ours but assumes the opposite extreme, namely that PRDM9 binding is non-limiting, and their qualitative results are similar to ours. However, while the degree of competitive binding does not appear to affect the qualitative results, it is will affect quantitative predictions. We therefore focus on the

qualitative insights about PRDM9 evolution and the interpretation of empirical observations gleaned from our simple model.

## The erosion of the strongest hotspots drives PRDM9 evolution

Gene conversion acts like selection against PRDM9 binding sites, leading to their loss over time (*Myers et al., 2010*). In our two-heat model, we assume the selection on hotspots is stronger than considered in a previous model ($s \approx 0.006$ to $0.021$, corresponding to the range of dissociation constants considered, compared to $s \approx 0.003$; *Latrille et al., 2017*), although far weaker than inferred from the under-transmission of exceptionally hot alleles within humans ($s \approx 0.46$ to $0.53$; *Jeffreys and Neumann, 2002*; *Berg et al., 2011*). A recent estimate of the strength of drive across PRDM9 binding sites suggested that selection may in fact be substantially weaker even than previous models assumed ($s \approx 5 \times 10^{-5}$ to $2 \times 10^{-4}$) and comparable in strength to genetic drift (*Spence and Song, 2019*). If so, drive would be too weak to explain the turnover of PRDM9 in terms of previous models. In light of these results, *Spence and Song, 2019* suggested that only a small subset of strong PRDM9 binding sites, subject to much stronger drive than most, is crucial for the proper segregation of chromosomes during meiosis and it is their erosion that drives PRDM9 evolution. The findings of our two-heat model are consistent with their conjecture: both the rate of DSB formation and the probability of being bound symmetrically by PRDM9 are determined by the underlying binding affinity of a given site. Accordingly, the sites most likely to be symmetrically bound will also experience the highest rates of drive, in accordance with empirical observations (*Hinch et al., 2019*). We therefore suggest that the hotspots in the two-heat model can be thought of as approximating the behavior of the strongest PRDM9 binding sites within a more realistic, continuous distribution of binding affinities.

## Does the erosion of hotspots lead to more or fewer hotspots?

In our two-heat model, when hotspots are lost, PRDM9 binding shifts towards the more plentiful class of weak binding sites. Given a more realistic distribution of heats, we would similarly expect the loss of hotspots to result in PRDM9 binding becoming more dispersed across the genome. Consider, for example, a crude approximation in which PRDM9 binding is strongest at loci with the optimal binding motif, with $n$ non-degenerate bases, binding is weaker by comparison at $\sim 3n$ times as many motifs with one mismatch, and so forth. After the optimal binding motifs have eroded, most PRDM9 binding will plausibly shifts toward the next best and larger class of motifs, with one mismatch, and then those with two mismatches, and so on. This consideration suggests that the number of hotspots will increase over time (although the extent to which binding becomes more dispersed will depend on the degree to which PRDM9 binding is limiting). This picture is in stark contrast with the previous understanding of the evolution of hotspots, namely, that gene conversion leads to a reduction in the number of hotspots over time. It further suggests that the heat of observed hotspots should decrease over time. Taken to the extreme, in the absence of PRDM9 turnover, the number of hotspots would continue to increase until the point at which they are no longer hot enough to be considered hotspots.

The evolution of hotspots can also be viewed from other perspectives. The standard, operational definition of hotspots is as short intervals of the genome (typically <2 kilobases) with recombination rates that exceed the local or genome-wide background rate by more than some arbitrary multiplicative factor (e.g., fivefold; reviewed in *Tock and Henderson, 2018*). By such a definition, whether or not the number of hotspots is actually increasing or decreasing over time as a consequence of gene conversion will depend on the threshold used to define them. Another measure of the number of hotspots, which is arguably more relevant biologically and less arbitrary, could be based on the average number of symmetrically bound sites. From this perspective, we expect the number of hotspots to decrease so long as the probability of being symmetrically bound increases with the probability of experiencing gene conversion.

Previous models suggested that hotspot evolution can be described as a *Red Queen* theory dynamic, referencing the Red Queen's (in Lewis Carroll's *Through the Looking-Glass*) statement that "it takes all the running you can do, to keep in the same place." In these models, the number of hotspots associated with a given PRDM9 allele is constantly decreases, and PRDM9 evolves rapidly ('runs') to keep the number of hotspots roughly constant. In our model, the queen is still running to stay in the same place, but the direction in which she is running is a matter of perspective. We liken this to the optical illusion of the Penrose stairs. Walking around the stairs in one direction, there is a gradual

decrease in the number of symmetrically-bound sites over time, that is suddenly increased upon the appearance of a new PRDM9 allele. Walking in the other direction, there is a gradual increase in the number of loci that are used for recombination, that is suddenly decreased upon the appearance of a new PRDM9 allele. Either way you go around, however, you end up in the same place.

## Diversity of the PRDM9 ZF-array

In all species with a complete PRDM9 investigated to date, PRDM9 diversity appears to be high relative to other ZF genes, regardless of the effective population size (*Buard et al., 2014*; *Vara et al., 2019*; *Alleva et al., 2021*). The rate of mutation at the PRDM9 ZF array has been inferred to be quite high, with a ZF copy number change of ~0.3–20 x $10^{-5}$ per generation (*Jeffreys et al., 2013*), as expected from its minisatellite nature. The high mutation rate, however, may be counterbalanced by strong selection favoring specific PRDM9 ZF alleles, as it is for other ZF genes (*Ubeda and Wilkins, 2011*; *Latrille et al., 2017*; this study).

In our two-heat model, PRDM9 diversity depends primarily on the population size and the heat of the hotspots considered. With weak hotspots, we observe a cycle in which one PRDM9 allele dominates the population at a time. Diversity is therefore relatively low throughout most of the cycle, but peaks when the dominant allele becomes susceptible to invasion and is eventually lost, as new alleles compete for fixation. With strong hotspots and large population sizes, new PRDM9 alleles constantly invade before any can fix, resulting in a consistently high level of diversity. With strong hotspots and small population sizes, there are intermittent, chaotic cycles driven by an occasional fixation, with periods of high diversity—resembling those in larger populations—in between.

It is unclear which of these dynamics is closer to the one in nature, namely given the true distributions of binding affinities. The strong signatures of erosion at binding sites associated with PRDM9 alleles in several species (*Myers et al., 2010*; *Smagulova et al., 2016*; *Hoge et al., 2023*) seem indicative of a cycling dynamic. In turn, anecdotal evidence suggests that being heterozygote for a minor PRDM9 ZFs is protective against azoospermia in humans (*Irie et al., 2009*), and the high levels of PRDM9 diversity seen across species with very different effective population sizes might suggest that the dynamics are closer to expectations for strong hotspots. A better understanding of PRDM9 dynamics, and diversity in particular, would be aided by more systematic surveys of diversity levels across species, alongside signatures of lineage-specific erosion and tests for different modes of selection at the PRDM9 locus.

In these regards, it would also be useful to keep in mind a couple of additional factors. First, we assume a panmictic population, whereas population structure is expected to increase diversity. Second, we assume that each new PRDM9 allele recognizes an entirely new set of binding sites, whereas PRDM9 alleles often share a substantial degree of overlap of binding sites (such as for the PRDM9A and PRDM9B alleles in humans) (*Pratto et al., 2014*; *Altemose et al., 2017*). Consequently, the number of effectively distinct alleles may be smaller than the number defined based of their observed ZFs.

## PRDM9-mediated hybrid sterility

Most empirical manipulations that rescue fertility in otherwise sterile crosses of hybrid mice can be readily understood as restoring symmetric binding (*Davies et al., 2016*; *Gregorova et al., 2018*; *Davies et al., 2021*). Two of these manipulations require more nuanced explanations, however: notably, why is it that fertility can be restored by either (i) increasing the expression of either PRDM9 allele present (*Flachs et al., 2012*), or by (ii) removing one allele but not the other, for example removing the B6 allele in a cross between B6 and PWD mice (*Flachs et al., 2012*). Considering these findings in light of our model suggests a plausible explanation.

In the sterile hybrids, no sites experience a symmetric DSB on the smallest chromosome. Nonetheless, binding sites for both alleles occur on their respective parental backgrounds, as evidenced by their use in the parental mice. Why then are they not used in the hybrids? We suggest that the reason is because PRDM9 is limiting, and the available PRDM9 is being sequestered to the non-parental genetic background, where their strongest binding sites have not been eroded. Increasing PRDM9 expression of either allele results in more PRDM9 binding at parental binding sites, allowing for symmetric binding. In turn, the effect of removing a PRDM9 allele should depend on which allele exhibits more symmetric binding in the hybrid. When one allele is removed, a greater number of DSBs

localize to sites bound by the remaining allele. Consequently, removing the allele that exhibits more symmetric binding should further decrease the probability of a symmetric DSB, whereas removing the other allele should increase it.

## Why use hotspots?

Given recent findings, the evolutionary benefit of PRDM9 binding symmetrically plausibly stems from the preferentially use of the same loci on one chromosome for DSB initiation and, on the homologous chromosome, for DSB repair (see Introduction). This strategy improves the efficacy of both homolog pairing and DSB repair, potentially by reducing the search space during homolog engagement and/ or by increasing the accessibility of the loci for strand invasion and subsequent steps of repair. It may further improve the accuracy of the repair process by reducing the odds of ectopic exchange. *Davies et al., 2016* hypothesized that by the same reasoning, the advantage of hotspots may be to limit recombination to a small proportion of the genome (in species that do not use PRDM9 but have hotspots, such as birds, as well as those that do; *Singhal et al., 2015*). In this regard, it is noteworthy that taxa with DSB-independent mechanisms of homolog pairing, such as *Drosophila* or *C. elegans*, do not have hotspots (*Manzano-Winkler et al., 2013*; *Smukowski Heil et al., 2015*; *Kaur and Rockman, 2014*; *Bernstein and Rockman, 2016*), whereas most species with DSB-dependent mechanisms, such as most yeast, plants or vertebrates, do (*Auton et al., 2013*; *Singhal et al., 2015*; *Lam and Keeney, 2015*; *Yelina et al., 2015*; *Tock and Henderson, 2018*; *Liu et al., 2019*).

In our model, as well as in a more recent model from *Genestier et al., 2023*, the benefit of symmetric PRDM9 binding is assumed from the outset, by postulating that fitness is a function of the probability of a symmetric DSB. In both models, the erosion of the strongest hotspots during the lifespan of a PRDM9 allele leads to a smaller proportion of symmetrically bound hotspots, and given the fixed and limiting number of DSB in both models, results in fewer symmetric DSB and lower fitness. Our assumption that PRDM9 binding is limiting leads to a further benefit of maintaining a small number of strong hotspots, because when there are too many, the competition among strong hotspots for PRDM9 makes symmetric binding even less likely. One could envision future extensions of our model in which the benefits of symmetric DSB arise mechanistically rather than being postulated, by modeling of the processes of homolog pairing and/or DSB repair (e.g. in which the benefit of symmetry is reduced in the presence of excess PRDM9 binding, because PRDM9 binding becomes less indicative of homology). Nonetheless, even without such extensions, our results clarify how the requirement for symmetry shapes the recombination landscape by favoring the use of a small proportion of the genome for recombination.

## Why use PRDM9?

PRDM9 arose prior to the common ancestor of animals and has since been repeatedly lost, including more than a dozen times across vertebrates (*Baker et al., 2017*; *Cavassim et al., 2022*). In vertebrate lineages in which PRDM9 has been lost, such as canids, birds or percomorph fish, as well as in PRDM9 knockout mice and rats, and to a minor degree in sterile mice hybrids, recombination events are concentrated in hotspots that tend to be found associated with promoter-like features of the genome, such as transcriptional start sites and/or CpG-islands (*Brick et al., 2012*; *Auton et al., 2013*; *Singhal et al., 2015*; *Smagulova et al., 2016*; *Mihola et al., 2021*). The same is true for lineages that are not believed to have ever carried PRDM9 but have recombination hotspots, such as in plants or fungi (*Yelina et al., 2015*; *Lam and Keeney, 2015*). Therefore, this mechanism for localizing hotspots is likely ancestral to the evolution of PRDM9's role in recombination, and is retained in the presence of PRDM9. Indeed, recent work suggests that the two mechanisms may be engaged in a tug of war, the outcome of which is more lopsided in mice and humans and more even in other species, such as snakes (*Hoge et al., 2023*).

Given the existence and retention of an alternative mechanism, the benefit of PRDM9 is unclear. If species without PRDM9 also recruit factors involved in both DSB initiation and DSB repair on both homologs, then PRDM9 may be favored when it further enhances the degree of symmetric recruitment of such factors relative to that of default hotspots, that is when PRDM9-mediated hotspots are fewer and hotter than default hotspots given the same number of DSBs. In support of this possibility, DSB heats measured by DMC1 ChIP-seq are higher in wild type than in PRDM9-/- mice (*Brick et al., 2012*).

If PRDM9 improves the efficiency of DSB-dependent homolog engagement, it may also enable the successful completion of meiosis with fewer DSBs—a potentially important secondary benefit, given that DSBs pose an inherent danger to cellular survival (*Cooper et al., 2016*). While the number of DSBs formed per meiosis is regulated independently of PRDM9, we speculate that, given a requirement for a certain number of 'symmetric DSBs' per meiosis, the number of DSBs made in a typical germ cell might co-evolve with the degree of symmetric PRDM9 binding. Notably, not all strains of mice are sterile when PRDM9 is knocked out, and those which are tend to have fewer DSBs per meiosis (*Mihola et al., 2019*), which seems to support the hypothesis that PRDM9 may be especially important for the successful completion of meiosis with fewer DSBs.

Another evolutionary puzzle remains open: why PRDM9 has been lost so many times in different lineages (*Baker et al., 2017*; *Cavassim et al., 2022*). Perhaps the answer to this question also relates to the advantage of symmetry. In species in which segregating PRDM9 alleles no longer provide an advantage over the default use of promoter-like features, there may no longer be selection to maintain PRDM9, allowing it to be lost by genetic drift. Alternatively, the reduced symmetry at default sites relative to PRDM9 binding sites might be compensated for by larger numbers of DSBs (possibly present, despite their inherently dangerous nature, for other evolutionary reasons), or because other changes increase the efficiency of DSB repair, rendering symmetry less important.

## Acknowledgements

We thank Nicolas Lartillot, Laurent Duret, Scott Keeney, and members of the Przeworski and Sella labs for helpful discussions, as well as our reviewers, including Bernard de Massy and Sylvain Glemin, for their useful comments. This work was supported by NIH grant R01 GM83098 to MP and NIH R01 GM115889 to GS.

## Additional information

### Funding

| Funder | Grant reference number | Author |
| --- | --- | --- |
| National Institutes of Health | R01 GM83098 | Molly Przeworski |
| National Institutes of Health | R01 GM115889 | Guy Sella |

The funders had no role in study design, data collection and interpretation, or the decision to submit the work for publication.

### Author contributions

Zachary Baker, Conceptualization, Formal analysis, Investigation, Methodology, Writing – original draft, Writing – review and editing; Molly Przeworski, Conceptualization, Supervision, Funding acquisition, Methodology, Writing – original draft, Writing – review and editing; Guy Sella, Conceptualization, Formal analysis, Supervision, Funding acquisition, Investigation, Methodology, Writing – original draft, Writing – review and editing

### Author ORCIDs

Zachary Baker ⓘ https://orcid.org/0000-0002-1540-0731
Molly Przeworski ⓘ https://orcid.org/0000-0002-5369-9009
Guy Sella ⓘ https://orcid.org/0000-0002-5239-7930

### Decision letter and Author response

Decision letter https://doi.org/10.7554/eLife.83769.sa1
Author response https://doi.org/10.7554/eLife.83769.sa2

## Additional files

### Supplementary files

### Data availability

All modeling code, as well as code used to generate the figures, is available at https://github.com/sellalab/PRDM9_model (copy archived at *Sellalab, 2023*). Source data files have been provided for Figures 2–6 and their associated figure supplements, as well as for the figures in Appendices 4–5.

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

## Appendix 1

### Molecular dynamics of PRDM9 binding

We first consider the case in which all binding sites for a given PRDM9 allele have the same binding affinity. At equilibrium, the rate at which PRDM9 molecules dissociate from their binding sites equals that rate at which they bind them. Denoting the expected number of bound PRDM9 molecules by $P_B$ and the rate of dissociation of a bound PRDM9 molecule by $k_{off}$, the total rate of PRDM9 dissociation is $k_{off} \cdot P_B$ (see *Table 1* for a summary of notation). Further denoting the expected number of free PRDM9 molecules by $P_F$, the expected number of free binding sites by $n_F$ and the rate of association between a free PRDM9 molecule and a given binding site by $k_{on}$, the total rate at which PRDM9 molecules bind is $k_{on} \cdot n_F \cdot P_F$. Therefore, at equilibrium

$$k_{off} \cdot P_B = k_{on} \cdot n_F \cdot P_F$$

or equivalently

$$k \cdot P_B = n_F \cdot P_F \tag{S1.1}$$

where $k \equiv k_{off}/k_{on}$ is the dissociation constant. Substituting $n_F = 4n - P_B$ (where $n$ is the total number of binding sites *per chromatid*) into *Equation S1.1* and solving for the expected number of bound PRDM9 molecules, we find that

$$P_B = \frac{4n \cdot P_F}{P_F + k} \tag{S1.2}$$

In this way, we see that the probability that a given site is bound by PRDM9, its "heat", is

$$H = \frac{P_B}{4n} = \frac{P_F}{P_F + k} \tag{S1.3}$$

Next, we solve for the expected number of free PRDM9 molecules, $P_F$. Substituting $P_B = P_T - P_F$ (where $P_T$ is the total number of PRDM9 molecules) into *Equation S1.2* and rearranging it, we end up with a quadratic in $P_F$ :

$$P_F^2 + \left(k + 4n - P_T\right)P_F - kP_T = 0$$

The general solution for this quadratic is

$$P_F = \frac{\pm\sqrt{\left(k + 4n - P_T\right)^2 + 4kP_T} - \left(k + 4n - P_T\right)}{2},$$

but further noting that $4kP_T > 0$ and that the expected number of free PRDM9 molecules must be positive, we find that

$$P_F = \frac{\sqrt{\left(k + 4n - P_T\right)^2 + 4kP_T} - \left(k + 4n - P_T\right)}{2}. \tag{S1.4}$$

The case in which a PRDM9 molecule has binding sites with different affinities can be treated analogously. In this case, the equilibrium requirement must be met for sites of each affinity. By analogy to *Equation S1.2*, we find that

$$P_B^i = \frac{P_F \cdot 4n_i}{P_F + k_i}. \tag{S1.5}$$

where $P_B^i$ is the expected number of PRDM9 molecules bound to sites with the $i$-th affinity, $n_i$ is the total number of these sites and $k_i$ is their dissociation constant. Similarly, by analogy to *Equation S1.3*, we find that the probability that a binding site with the $i$-th affinity is bound by PRDM9, its "heat", is

$$H_i = \frac{P_B^i}{4n_i} = \frac{P_F}{P_F + k_i}. \tag{S1.6}$$

In order to solve for the expected number of free PRDM9 molecules, we note that

$$P_F = P_T - \sum_{i=1}^{m} P_B^i,$$

where $m$ is the number of different binding affinities. Substituting the expression for $P_B^i$ from *Equation S1.5* into this one, we find that

$$P_F = P_T - \sum_{i=1}^{m} \frac{P_F \cdot 4n_i}{P_F + k_i} \qquad (S1.7)$$

which is a polynomial of degree $m + 1$ in $P_F$ that can be solved numerically.

In the case with two binding affinities, rearranging *Equation S1.7* yields the cubic equation:

$$P_F^3 + b \cdot P_F^2 + c \cdot P_F + d = 0$$

with coefficients $b = \left(4n_1 + 4n_2 + k_1 + k_2 - P_T\right)$ , $c = \left(4k_1 n_2 + 4k_2 n_1 + k_1 k_2 - \left(k_1 + k_2\right) P_T\right)$ and $d = -k_2 k_1 P_T$ . The solution of this equation is given by

$$P_F = \sqrt[3]{R + \sqrt{Q^3 + R^2}} + \sqrt[3]{R - \sqrt{Q^3 + R^2}} - b/3$$

where $R = \left(9bc - 27d - 2b^3\right)/54$ and $Q = \left(3c - b^2\right)/9$ .

## Appendix 2

### The probability of a symmetric DSB

We define a "symmetric DSB" at a given binding site as the event in which, considering the four chromatids, at least one of four copies of the binding site (across the four copies of the chromosome present) during meiosis has been bound by PRDM9 and experienced a DSB and at least one of the two copies on the homologous chromosomes has also been bound by PRDM9.

We first consider the probability that at least one of the two copies on a pair of sister chromatids has been bound by PRDM9 and experienced a DSB. The probability that a given copy with binding affinity $\sim 3n$ has been bound is $H_i$ ; having been bound, the probability that it experiences a DSB is $c$, and therefore the probability of both is $cH_i$ . The probability that at least one of the two copies on sister chromatids experienced both is the complement of the probability that neither of them did, that is

$$1 - \left(1 - cH_i\right)^2 = cH_i \left(2 - cH_i\right)$$

Similarly, the probability that at least one of the copies on the homologous chromosomes has been bound by PRDM9 is

$$1 - \left(1 - H_i\right)^2 = H_i \left(2 - H_i\right)$$

Assuming independence, the probability that the binding site on a given homologous chromosome has been bound and experienced a DSB and that the corresponding site on the other homologous chromosome has been bound is the product of the above expressions

$$cH_i^2 \left(2 - H_i\right) \left(2 - cH_i\right)$$

Lastly, given that either chromosome could be the one to experience the DSB, the probability of a 'symmetric DSB' is the complement of the probability that a 'symmetric DSB' occurs in neither direction, that is

$$\alpha_i = 1 - \left(1 - cH_i^2 \left(2 - H_i\right) \left(2 - cH_i\right)\right)^2 \tag{S2.1}$$

By the same token, the probability that a site with binding affinity $i$ has been symmetrically bound, regardless of DSBs (as shown in *Figure 2B*), is the complement of the probability that that neither chromatid on either homolog has been bound, i.e.,

$$H_i^S = \left(1 - \left(1 - H_i\right)^2\right)^2 \tag{S2.2}$$

# Appendix 3

## The competitive effect of weak binding sites in the two-heat model

The molecular and evolutionary dynamics of a PRDM9 allele in the two-heat model depends on three parameters that describe allele properties and on two variables. The parameters are the number of PRDM9 molecules, $P_T$, the dissociation constant of hotspots, $k_1$, and the dissociation constant of weak binding sites, $k_2$; the variables are the total numbers of hotspots, $n_1$, and of weak binding sites, $n_2$. Here we show that in the parameter regime of interest to us, we can describe the dynamics in terms of a single variable ($n_T^1$ and two parameters: $P_T$ and $k_1/P_F^w$, where $P_F^w$ is the expected number of free PRDM9 molecules in the absence of hotspots. The latter, compound parameter reflects the competitive effect of the backdrop of weak binding sites on binding hotspots. This reduction in the number of parameters greatly simplifies our analysis in the main text.

We assume that there are a small number of strong binding sites, which are bound symmetrically at appreciable rates—termed "hotspots" in the main text—and a larger number of weaker binding sites, which compete with hotspots for PRDM9 binding, but are rarely bound symmetrically. Specifically, we require that weak binding sites are much weaker and more numerous than hotspots, i.e., $k_2 \gg k_1$ and $n_2 \gg n_1$ respectively, and that the number of weak binding sites far exceeds the number of PRDM9 molecules, i.e., $n_2 \gg P_T$, such that any given weak binding site is rarely bound, i.e., $H_2 \ll 1$. Alongside **Equation S1.6**, the latter condition implies that

$$H_2 \cong \frac{H_2}{1 - H_2} = \frac{P_F}{k_2} \ll 1, \tag{S3.1}$$

and thus that

$$k_2 \gg P_F. \tag{S3.2}$$

These conditions also imply that the change to the number of weak binding sites is negligible, implying that we can treat $n_2$ as approximately constant, leaving us with a single variable: the number of hotspots, $n_1$. We also require that most PRDM9 molecules are bound at equilibrium, i.e., that $P_F \ll P_T$. In its extreme, this implies that the expected number of free PRDM9 molecules in the absence of hotspots $P_F^w \ll P_T$.

In order to demonstrate that under these conditions, we can approximate the dynamics in terms of two parameters ($P_T$ and $k_1/P_F^w$), we show that the equations that govern the evolution of a PRDM9 allele and its binding sites are well approximated using these parameters alone. First, with negligible probability of a symmetric DSB at a weak binding site ($\alpha_2 \approx 0$), the fitness of a homozygote for a PRDM9 allele is well approximated by

$$W_{hom} = 1 - \left(1 - \alpha_1\right)^{rn_1} \left(1 - \alpha_2\right)^{rn_2} \approx 1 - \left(1 - \alpha_1\right)^{rn_1}, \tag{S3.3}$$

which depends only on $n_1$ and on the probability of a symmetric DSB at hotspots, $\alpha_1$. The same can be easily shown for the fitness of a heterozygotes for PRDM9: namely, that it depends only on $n_1$ and $\alpha_1$ of the two PRDM9 alleles. In turn, both the probability of a symmetric DSB at a hotspot (**Equation 4**):

$$\alpha_1 = 1 - \left(1 - cH_1^2 \left(2 - H_1\right) \left(2 - cH_1\right)\right)^2,$$

and the rate of gene conversion experienced by hotspots (**Equation 9**):

$$g_1 = BcH_1,$$

depend on the probability of hotspots being bound, $H_1$, and on the probability of bound sites experiencing a DSB, $c$. Given that $P_F \ll P_T$, the probability of bound sites experiencing a DSB is well approximated by

$$c = \frac{D}{P_B} = \frac{D}{P_T - P_F} \approx \frac{D}{P_T},$$

with $P_T$ being the total number of PRDM9 molecules in homozygotes (the result for heterozygotes for PRDM9 is the same, given our assumption that the sum of the number of PRDM9 molecules of

both kinds equals $P_T$ ; see **Equation 3**). Thus, what remains for us to show is that $H_1$ depends only on $n_1$ , $P_T$ and $k_1/P_F^w$ .

To this end, we derive an equation for $H_1$ and show that it can be expressed in terms of these parameters alone. First, we express the total number of PRDM9 molecules as the sum of those that are free, bound to hotspots, or bound to weak binding sites:

$$P_T = P_F + 4n_1 H_1 + 4n_2 H_2.$$

Substituting $H_2 \approx P_F/k_2$ (**Equation S3.1**) and $P_F = H_1 k_1/\left(1 - H_1\right)$ (from **Equation S1.6**) and rearranging this equation, we find that

$$\frac{4n_1}{P_T} H_1^2 - \left( 1 + \frac{4n_1}{P_T} + k_1 \frac{\left(1 + 4n_2/k_2\right)}{P_T} \right) H_1 + 1 \approx 0. \tag{S3.4}$$

Next, we solve for the number of free PRDM9 in the absence of hotspots. Once again, we express the total number of PRDM9 molecules as a sum of free and bound ones, and substitute $H_2 \approx P_F/k_2$ (**Equation S3.1**), where in this case, we find that

$$P_T = P_F^w + 4n_2 H_2^w \approx P_F^w + 4n_2 \frac{P_F^w}{k_2} = P_F^w \left(1 + 4n_2/k_2\right),$$

and thus that

$$P_F^w \approx \frac{P_T}{1 + 4n_2/k_2} . \tag{S3.5}$$

Substituting this expression into **Equation S3.4**, we attain a quadratic for $H_1$ ,

$$\frac{4n_1}{P_T} H_1^2 - \left( 1 + \frac{4n_1}{P_T} + \frac{k_1}{P_F^w} \right) H_1 + 1 \approx 0, \tag{S3.6}$$

with coefficients that are expressed in terms of $n_T^1$ , $P_T$ and $k_1/P_F^w$ .

## Appendix 4

### The effect of PRDM9 expression level

As we noted, there is considerable uncertainty about the number of PRDM9 molecules expressed during meiosis (see section on *Estimated parameters*). In the result section of the main text, we assumed this number to be 5000, based on empirical estimates of the number of PRDM9-bound sites per meiosis in mice, on the assumption that that almost all expressed PRDM9 molecules are bound. In particular, our choice is roughly consistent with cytological observations of thousands of PRDM9 foci in meiotic nuclei in mice (*Parvanov et al., 2017*; *Grey et al., 2017*). It is also consistent with estimates of 4700±400 PRDM9-modified sites in B6 mice meiotic nuclei (*Baker et al., 2014*). To arrive at these estimates, Baker et al. divided H3K4me3 ChIP-seq reads at PRDM9-dependent peaks by MNase-digested input DNA. This approach plausibly overestimates the number of PRDM9-bound sites, however, because binding sites that were not bound by PRDM9 and should have been included in the denominator might have been digested by the MNase. Here, we therefore examine how our main results are affected by assuming smaller numbers of expressed PRDM9 molecules. Specifically, we set the number of PRDM9 molecules ($P_T$) to 500, 1000, or 2500.

Within this range, the qualitative dynamics of the two-heat model are insensitive to the level of PRDM9 expression level (compare *Appendix 4—figure 1* to *Figure 4*). In particular, the qualitative behaviors observed when varying the strength of hotspots and population size remain the same.

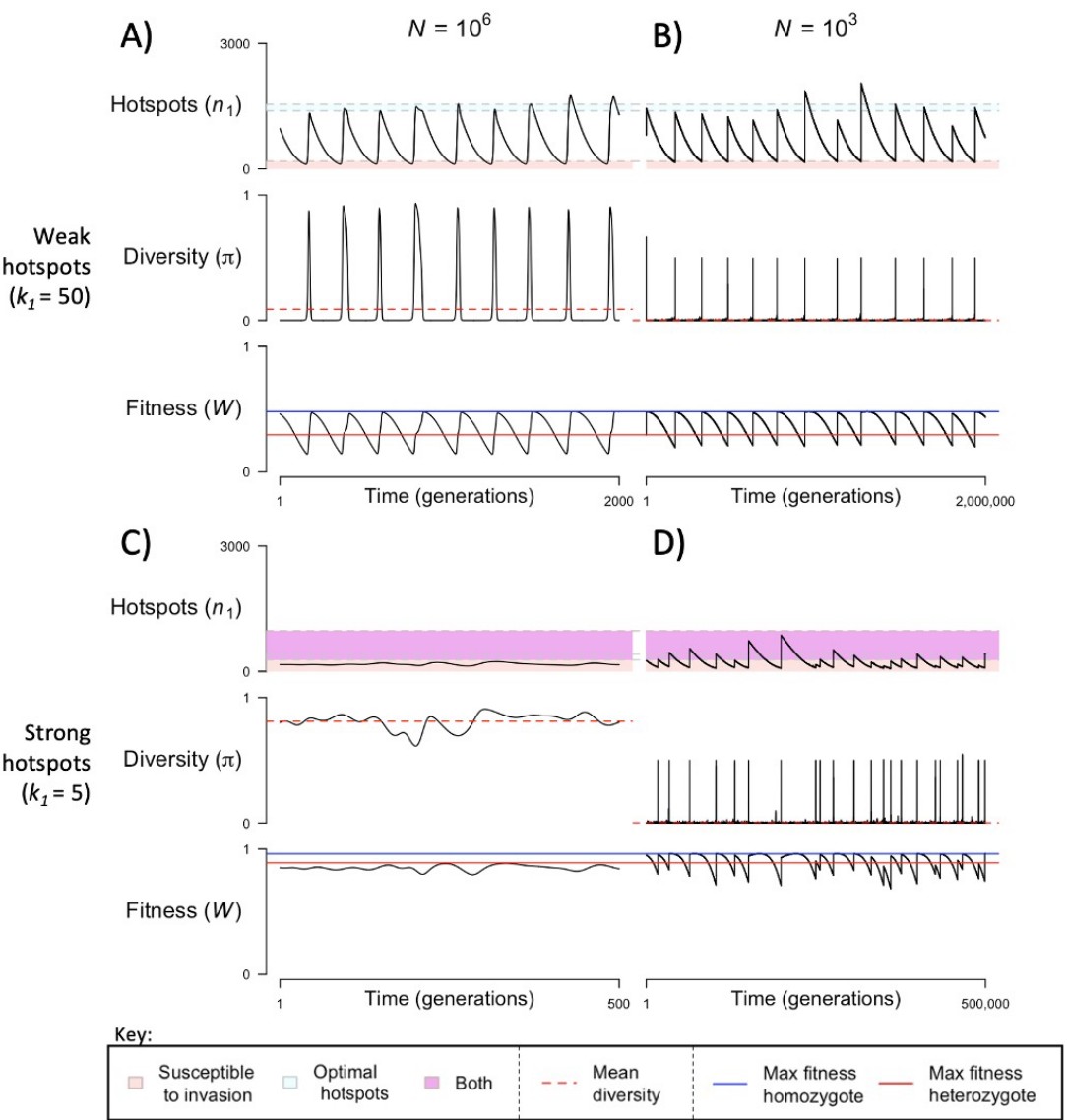

**Appendix 4—figure 1.** Dynamics of the two-heat model when $P_T = 1000$. The mean number of hotspots ($n_1$), diversity at PRDM9 ($\pi$), and mean fitness ($W$), as a function of time. We show four cases: with weak and strong hotspots (top and bottom rows respectively) and large and small population size (left and right column respectively). See *Figure 4* and Appendix 4 for details. This figure was generated using *Appendix 4—figure 1—source data 1–4*.

The online version of this article includes the following source data for appendix 4—figure 1:

**Appendix 4—figure 1—source data 1.** The number of hotspots ($n_1$), average population fitness ($W$), and diversity at the PRDM9 locus (pi) per generation, across 2,000 simulated generations of the two-heat model with a large population size ($N = 10^6$) and weak hotspots ($K_1 = 50$), and a smaller number of expressed PRDM9 molecules ($P_T = 1000$).

**Appendix 4—figure 1—source data 2.** The number of hotspots ($n_1$), average population fitness ($W$), and diversity at the PRDM9 locus ($\pi$) per generation, across 2,000,000 simulated generations of the two-heat model with a small population size ($N = 10^3$) and weak hotspots ($k_1 = 50$), and a smaller number of expressed PRDM9 molecules ($P_T = 1000$).

**Appendix 4—figure 1—source data 3.** The number of hotspots ($n_1$), average population fitness ($W$), and diversity at the PRDM9 locus ($\pi$) per generation, across 500 simulated generations of the two-heat model with a large population size ($N = 10^6$) and strong hotspots ($k_1 = 5$), and a smaller number of expressed PRDM9 molecules ($P_T = 1000$).

**Appendix 4—figure 1—source data 4.** The number of hotspots ($n_1$), average population fitness ($W$), and diversity at the PRDM9 locus ($\pi$) per generation, across 500,000 simulated generations of the two-heat model with

a small population size ($N = 10^3$) and strong hotspots ($k_1 = 5$), and a smaller number of expressed PRDM9 molecules ($P_T = 1000$).

The effects of PRDM9 expression level on key quantities can be largely understood in terms of its effect on the optimal number of hotspots (*Appendix 4—figure 2*), which decreases with PRDM9 expression level ($P_T$ in both homozygotes and heterozygotes. Specifically, the optimal number in homozygotes approaches $P_T/4$ as the heat of hotspots increases, or equivalently, as $k_1$ approaches zero (see *Appendix 4—figure 2A*). Consequently, invading PRDM9 alleles have fewer hotspots when PRDM9 expression is lower (*Appendix 4—figure 2B*). By the same token, established PRDM9 alleles become susceptible to invasion, and thus exit the population with fewer hotspots, when PRDM9 expression is lower (*Appendix 4—figure 2B*).

The effect of PRDM9 expression level on turnover time is mediated by the changes to the number of hotspots of invading and exiting PRDM9 alleles, with the effect approximately canceling out in the case of weak hotspots ($k_1 = 50$) and resulting in a slight increase with increased expression level in the case of strong hotspots ($k_1 = 5$) (*Appendix 4—figure 2C*). In principle, we would expect diversity levels to increase when the turnover time decreases, which is not what we observe (*Appendix 4—figure 2C, D*). Instead, we observe an increase in diversity with expression, as a consequence of having the number of hotspots of newly arising PRDM9 alleles drawn from a uniform distribution between 1 and 5,000. Namely, lower expression levels lead to a lower number of hotspots associated with the PRDM9 alleles capable of invading, which results in an effectively lower mutation rate at PRDM9 and thus in lower diversity levels.

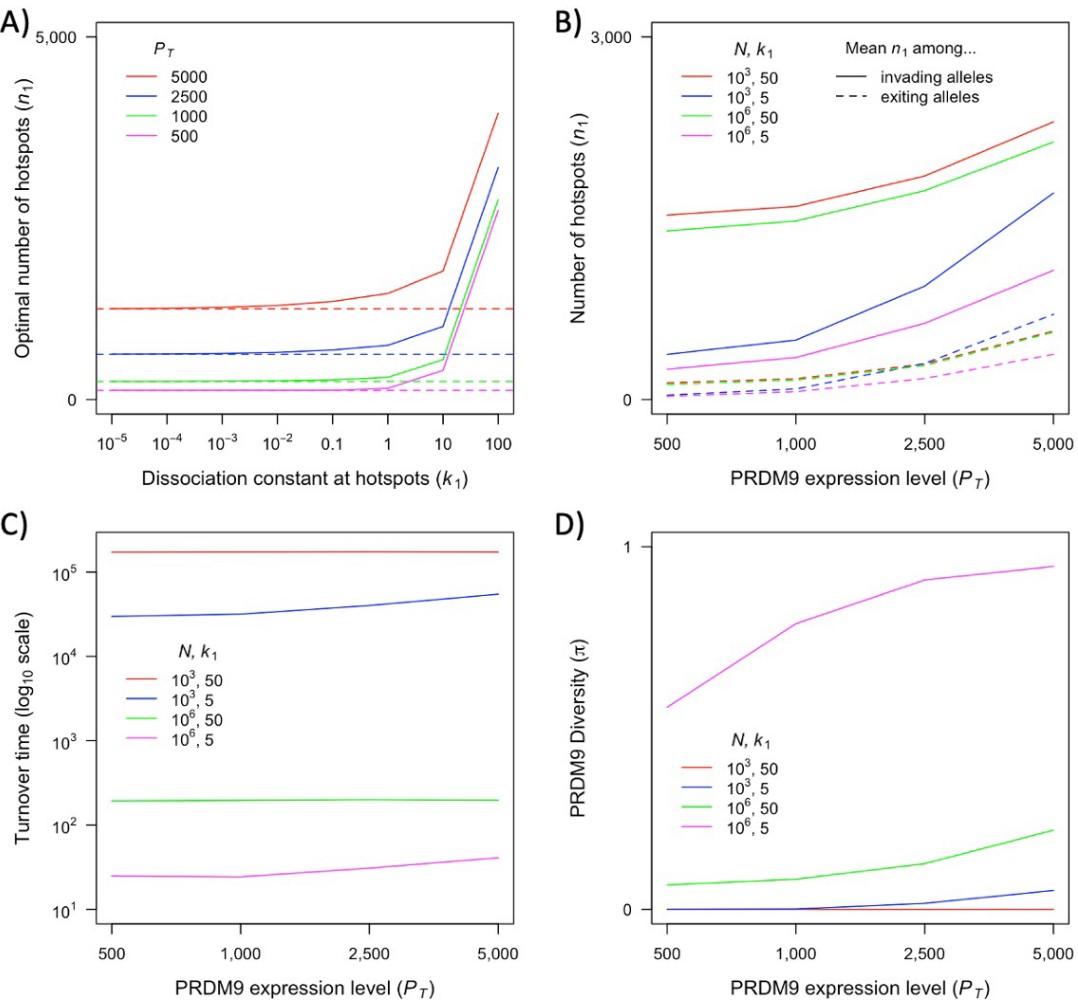

**Appendix 4—figure 2.** The dependence of key quantities on PRDM9 expression level. (**A**) The number of hotspots which optimizes fitness for an individual homozygous for PRDM9 alleles as a function of the dissociation constant at hotspots ($k_1$) for different levels of PRDM9 expression ($P_T$). Horizontal dashed lines indicate the expected optimal values as $k_1$ approaches zero ($P_T/4$). (**B**) Mean number of hotspots among incoming and exiting alleles in small or large population sizes ($N=10^3$ or $10^6$) and when hotspots are relatively strong or weak ($k_1 = 5$ or 50), as a function of PRDM9 expression level ($P_T$). (**C**) Average turnover time of segregating PRDM9 alleles for different parameters as a function of PRDM9 expression level ($P_T$). (**D**) Mean diversity at PRDM9 over time for different parameters as a function of PRDM9 expression level ($P_T$). The weighting of PRDM9 alleles in B and C is as detailed in the main text. This figure was generated using **Appendix 4—figure 2—source data 1**.

The online version of this article includes the following source data for appendix 4—figure 2:

**Appendix 4—figure 2—source data 1.** Summary statistics from simulations for large and small population sizes ($N = 10^3$ or $10^6$) and for small and large hotspot dissociation constants ($k_1 = 5$ or 50), for a range of values of the expression level of PRDM9 ($P_T = 500, 1000, 2500$ or $5000$), including mean turnover times, mean initial numbers of hotspots, and mean number of hotspots for exiting PRDM9 alleles (each weighted by the sojourn time of PRDM9 alleles), as well as the mean and interquartile range of diversity at PRDM9 ($\pi$). This data was used to generate **Appendix 4—figure 2**.

## Appendix 5

### The effect of the distribution of the initial number of hotspots

When we analyzed the dynamics of the 'two-heat model' in the main text, we assumed that the number of hotspots associated with new PRDM9 alleles are drawn from a uniform distribution across a wide range ($n_1 = 1 - 5000$), which spans the optimal values for both homozygotes and heterozygotes across the considered range of hotspot strengths ($k_1 = 5 - 50$). Our rationale was that under this assumption, the number of hotspots of PRDM9 alleles that persist and sometimes fix in the population will reflect the evolutionary dynamics rather than the properties of the mutational distribution. An alternative assumption, which is at the other extreme, would be to have the initial number of hotspots tightly distributed around a specific number of binding sites, namely one that arises from the specificity of a typical PRDM9 ZF array. Here, we briefly explore how making this alternative assumption would affect the dynamics.

We consider two such cases of 'tight' distributions, based on a rough approximation to the number of binding sites that perfectly matches the binding motif of a PRDM9 ZF array with $m = 10$ and 11 non-degenerate base pairs. To this end, we assume a diploid genome of size of $L = 1.75$ Gbp in which the nucleotide at each position is equally likely to be G, C, T and A. Under these assumptions, we would expect the number of hotspots associated with a new PRDM9 allele, $n_1$, to be Binomially distributed with $n_1 \sim B\left(2L, \left(1/4\right)^m\right)$. With $m = 10$, the distribution has mean ~3,338 and SD ~58, and with $m = 11$, it has mean ~834 and SD ~29.

The evolutionary dynamics in these cases strongly depend on whether individuals that are homozygote or heterozygote for the initial number of hotspots have higher fitness. First, we consider the case with $m = 10$ and mean ~3338,, in which the homozygotes have higher fitness throughout the range of hotspot strengths considered (*Figure 3*). This fact alone implies a cycling dynamic: a PRDM9 allele that invades is expected to fix because of the higher fitness in homozygotes, and to lose hotspots until it is invaded by the next allele. This is indeed what we observe in all cases (*Appendix 5—figure 1*). The higher fitness of new alleles in homozygotes precludes the 'fixed point' behaviors (*Figure 3B*) that give rise to highly polymorphic steady states or lead to a chaotic dynamic in the cases considered in the main text (*Figure 4C and D*, respectively). Instead, the dynamics we observe here are similar to what we see in the main text for weak hotspots ($k_1 = 50$), because in that case, selection favored new PRDM9 alleles with ~2,000 hotspots (*Figures 5 and 6A*), which have higher fitness in homozygotes than in homozygotes (compare *Figure 4A and B* with *Appendix 5—figure 1A, B*).

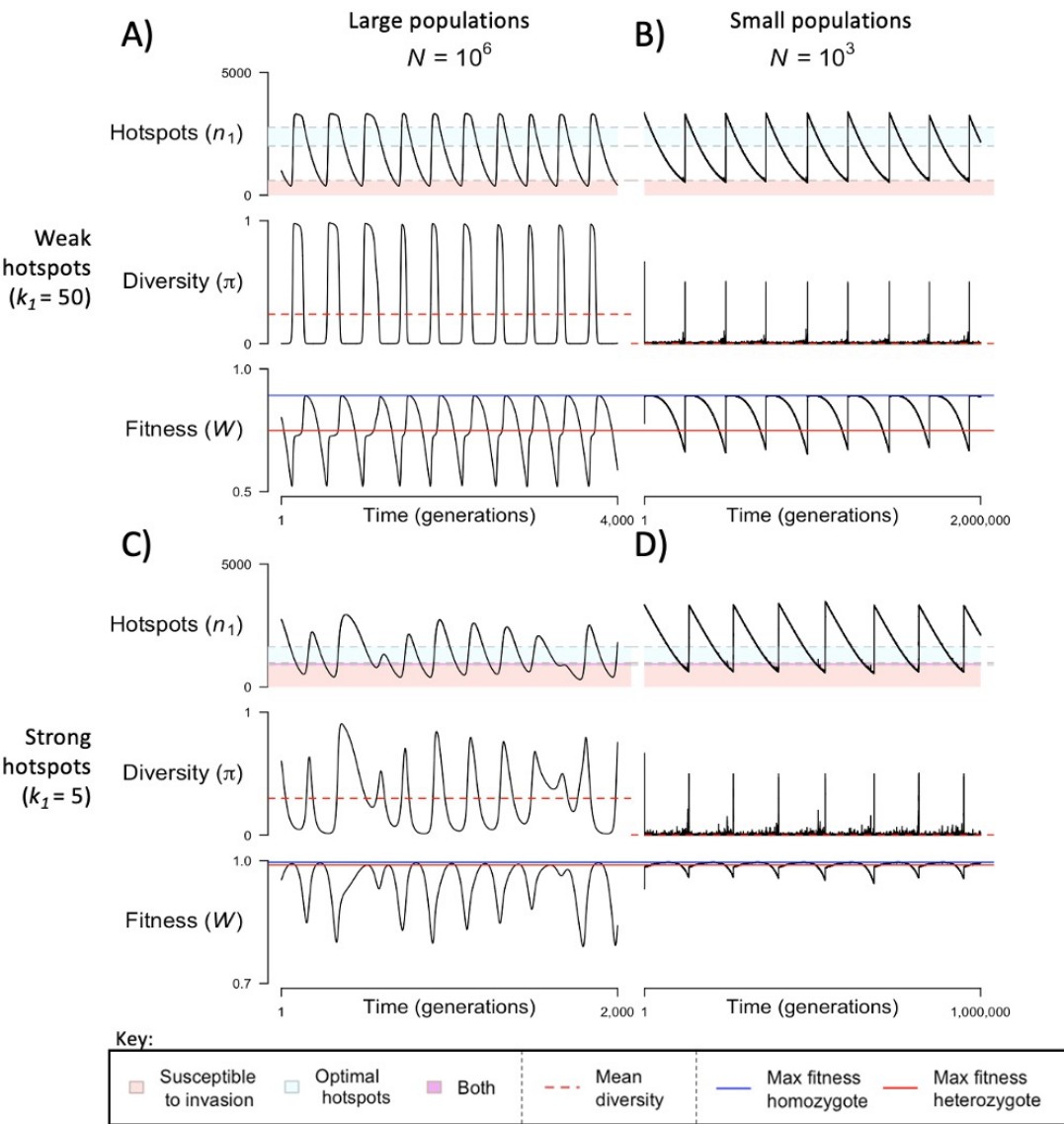

**Appendix 5—figure 1.** Dynamics of the two-heat model when the distribution of the initial numbers arises from a motif with 10 non-degenerate sites. The mean number of hotspots ($n_1$), diversity at PRDM9 ($\pi$), and mean fitness ($W$), as a function of time. We show four cases: with weak and strong hotspots (top and bottom rows respectively) and large and small population size (left and right column respectively). See *Figure 4* and Appendix 5 for details. This figure was generated using *Appendix 5—figure 1—source data 1–4*.

The online version of this article includes the following source data for appendix 5—figure 1:

**Appendix 5—figure 1—source data 1.** The number of hotspots ($n_1$), average population fitness ($W$), and diversity at the PRDM9 locus (pi) per generation, across 4,000 simulated generations of the two-heat model with a large population size ($N = 10^6$) and weak hotspots ($k_1 = 50$), when the distribution of the initial numbers arises from a motif with 10 non-degenerate sites.

**Appendix 5—figure 1—source data 2.** The number of hotspots ($n_1$), average population fitness ($W$), and diversity at the PRDM9 locus ($\pi$) per generation, across 2,000,000 simulated generations of the two-heat model with a small population size ($N = 10^3$) and weak hotspots ($k_1 = 50$), when the distribution of the initial numbers arises from a motif with 10 non-degenerate sites.

**Appendix 5—figure 1—source data 3.** The number of hotspots ($n_1$), average population fitness ($W$), and diversity at the PRDM9 locus ($\pi$) per generation, across 2,000 simulated generations of the two-heat model with a large population size ($N = 10^6$) and strong hotspots ($k_1 = 5$), when the distribution of the initial numbers arises from a motif with 10 non-degenerate sites.

**Appendix 5—figure 1—source data 4.** The number of hotspots ($n_1$), average population fitness ($W$), and diversity at the PRDM9 locus ($\pi$) per generation, across 1,000,000 simulated generations of the two-heat model

with a small population size ($N = 10^3$) and strong hotspots ($k_1 = 5$), when the distribution of the initial numbers arises from a motif with 10 non-degenerate sites.

Next, we consider the case with $n = 11$ and a mean of ~834 hotspots associated with new PRDM9 alleles. When hotspots are weak ($k_1 = 50$), homozygotes for new PRDM9 alleles have higher fitness than heterozygotes (*Figure 3A*). Consequently, we observe cycling dynamics, but in this case the turnover time is reduced, because it takes the loss of fewer hotspots for an established PRDM9 allele to become susceptible to invasion (compare *Figure 4A and B* with *Appendix 5—figure 2A, B*). In large populations ($N = 10^6$), in which hotspots erode faster than in smaller populations (*Equations 11 and 12*), an invading PRDM9 alleles becomes susceptible to invasion and selected against before it can even fix, resulting in more erratic, rapid cycles and high diversity levels (*Appendix 5—figure 2A*). The dynamics in *Appendix 5—figure 1C* can be explained in similar terms.

When hotspots are strong ($k_1 = 5$), heterozygotes for new PRDM9 alleles have higher fitness than homozygotes (*Figure 3B*). In large populations ($N = 10^6$), we observe the same fixed-point dynamic as reported in the main text: in that case, selection favored new PRDM9 alleles with ~1,070 hotspots (*Figures 5 and 6A*), which also have higher fitness in heterozygotes than in homozygotes (compare *Figure 4C* with *Appendix 5—figure 2C*). When the population size is small ($N = 10^3$), however, we do not observe the chaotic dynamics that we see in the main text (compare *Figure 4D* with *Appendix 5—figure 2D*). Here, new PRDM9 alleles never have a sufficient number of hotspots to allow them to persist and reach high frequencies in the population (and thus generate chaotic cycles). Instead, we observe a fixed-point dynamic that is similar to the dynamics in large populations, with new PRDM9 alleles constantly invading and exiting the population without reaching fixation, but in this case, the mutational input at PRDM9 is sufficiently low for there to be substantial stochastic variation in diversity levels (compare *Appendix 5—figure 2C, D*).

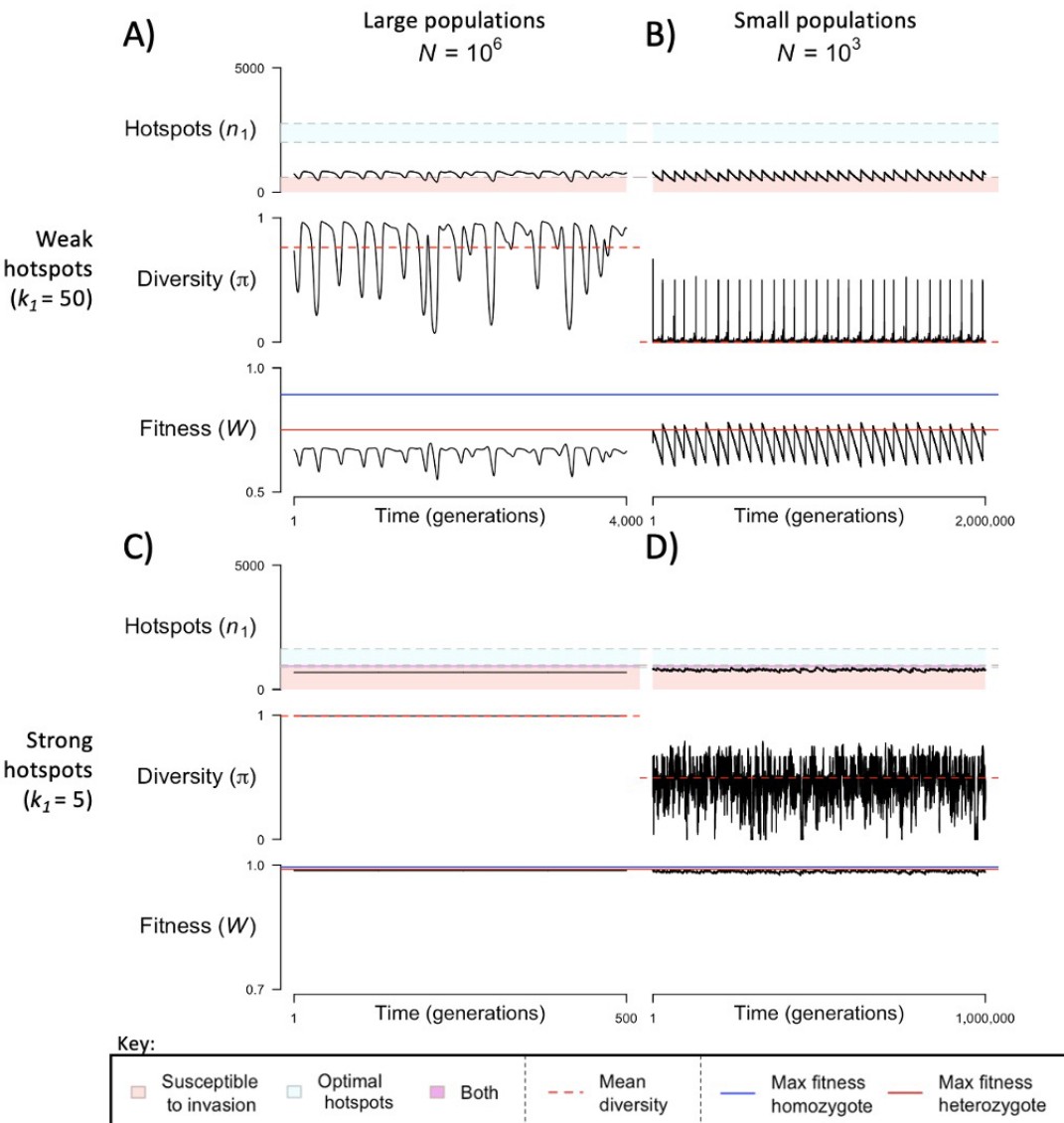

**Appendix 5—figure 2.** Dynamics of the two-heat model when the distribution of the initial numbers arises from a motif with 11 non-degenerate sites. The mean number of hotspots ($n_1$), diversity at PRDM9 ($\pi$), and mean fitness ($W$), as a function of time. We show four cases: with weak and strong hotspots (top and bottom rows respectively) and large and small population size (left and right column respectively). See **Figure 4** and Appendix 5 for details. This figure was generated using **Appendix 5—figure 2—source data 1–4**.

The online version of this article includes the following source data for appendix 5—figure 2:

**Appendix 5—figure 2—source data 1.** The number of hotspots ($n_1$), average population fitness ($W$), and diversity at the PRDM9 locus (pi) per generation, across 4,000 simulated generations of the two-heat model with a large population size ($N = 10^6$) and weak hotspots ($k_1 = 50$), when the distribution of the initial numbers arises from a motif with 11 non-degenerate sites.

**Appendix 5—figure 2—source data 2.** The number of hotspots ($n_1$), average population fitness ($W$), and diversity at the PRDM9 locus ($\pi$) per generation, across 2,000,000 simulated generations of the two-heat model with a small population size ($N = 10^3$) and weak hotspots ($k_1 = 50$), when the distribution of the initial numbers arises from a motif with 11 non-degenerate sites.

**Appendix 5—figure 2—source data 3.** The number of hotspots ($n_1$), average population fitness ($W$), and diversity at the PRDM9 locus ($\pi$) per generation, across 500 simulated generations of the two-heat model with a large population size ($N = 10^6$) and strong hotspots ($k_1 = 5$), when the distribution of the initial numbers arises from a motif with 11 non-degenerate sites.

**Appendix 5—figure 2—source data 4.** The number of hotspots ($n_1$), average population fitness ($W$), and diversity at the PRDM9 locus ($\pi$) per generation, across 1,000,000 simulated generations of the two-heat model

with a small population size ($N = 10^3$) and strong hotspots ($k_1 = 5$), when the distribution of the initial numbers arises from a motif with 11 non-degenerate sites.

