## [Editor Report]

This paper presents a theoretical model of the evolutionary dynamics of Prdm9-dependent meiotic recombination hotspots. This study provides important insights. It shows that selection acts to limit the number of hotspots and to increase hotspot symmetry. This is consistent with the proposed role of PRDM9 in coordinating DSB formation and repair. Although the authors did not explore all possible scenarios, the conclusions are convincing and open up directions for extending the model and testing some of its predictions.

---

## [Decision Letter]

**Decision letter after peer review:**

Thank you for submitting your article "Down the Penrose stairs: How selection for fewer recombination hotspots maintains their existence" for consideration by *eLife*. Your article has been reviewed by 3 peer reviewers, including Bernard de Massy as the Reviewing Editor and Reviewer #1, and the evaluation has been overseen by Jessica Tyler as the Senior Editor. The following individual involved in the review of your submission has agreed to reveal their identity: Sylvain Glemin (Reviewer #2).

Essential revisions:

1) Introduction could be reduced.

2) The objective of each step of the model is not clearly explained and it is left to the reader to understand where the authors want to go. At first read, it is not clear whether the authors present an analysis of the model or simulation results and why they do that. So, the results part deserves rewriting and re-organization to guide the reader.

3) The choice of key parameters is justified except for Prdm9 occupancy, and potential competition between different Prdm9 variants for DSB activity.

4) Developing the specific predictions relative to previous red-queen models, for Prdm9 diversity in human and mice, for gene dosage effects reported in mice.

5) Some more explanation about the difference between the Penrose stair and the Red-queen.

6) Several clarifications on the data presented in the figures are also required.

*Reviewer #1 (Recommendations for the authors):*

A few specific points also require clarification:

1) In the model, the question of competition is not entirely clear.

There may be different levels of competition:

­– Competition for binding of Prdm9 to its sites (this will depend on affinity and protein level);

Figure 1A indicates that Prdm9 binds its target as a consequence of competition between sites. It is not very clear what this means. Prdm9 will bind and occupy high-affinity sites more frequently than low-affinity sites. This is not a competition. There may be a competition but only if Prdm9 protein is limiting.

The authors write L215. "the corresponding PRDM9 binding sites will always be more likely to be bound in individuals homozygous for the cognate PRDM9 allele relative to heterozygotes": not if Prdm9 protein is not limiting.

In fact, by choosing conditions (L354) where 99% of Prdm9 is bound to weak sites in the absence of hotspots, it seems that the authors define conditions where Prdm9 is limiting. What is the rationale for this?

– Competition between Prdm9 bound and DSB activity (see Diagouraga et al., 2018) (if DSB activity is fixed, then the number of Prdm9 bound may exceed the number of potential DSB, ie 300). This could result in competition between Prdm9 alleles in heterozygotes.

2) Figure 1 represents k_on_ and Koff; the text refers to Ki, thus, even if obvious, it should be mentioned that ki=koff/kon.

3) Figure 3B: the area shaded in red should correspond for the context where the homozygous have lower fitness that the maximal fitness of the heterozygous. Thus, what is the red-shaded area for > 4800 hotspots?

4) Figure 5: The label of the y-axis is missing, please clarify. It would help to describe the figure in the text (L480-484) as way more consistent with the figure itself, and to better explain how the data from Figure 4 was used to derive these hotspot distributions.

I assume the initial and final hotspots correspond to exiting and invading alleles? Is this correct?

"…the distribution is centered close to the optimal number of hotspots.", I guess the authors refer to the invading alleles distribution, please clarify in the text.

5) Estimation of the number of Prdm9 bound sites.

The authors use the value that has been published by Baker et al. Genome Res 24, 724-32 (2014). However, the problem is that this number is questionable.

Baker et al. indicated:

"…we estimated that there are;4700 +/- 400 PRDM9-modified sites in the B6 background in an average meiosis." "For estimating the total number of PRDM9-dependent hotspots per meiosis, an equal volume of MNase-digested input DNA was analyzed alongside the ChIP DNA".

This method is not correct: by using this Mnase-digested input DNA, the amount of DNA is likely underestimated, as most input DNA molecules are not bound by Prdm9, do not have positioned nucleosomes at Prdm9 binding sites, and thus not protected from Mnase (depending on the duration of treatment) as positioned nucleosomes are. This unknown underestimation factor means that the number of sites bound by Prdm9 is overestimated.

The way to address this issue is to design models with different values for pT (from 5000 to 500 for instance). Does it affect the simulations? How?

6) Connections with in vivo data should be made:

Several important studies in the analysis of Prdm9 hybrid incompatibility have reported dosage effects (ie Mihola et al. Science 323, 373-5 (2009); Flachs, P. et al. PLoS Genet 8, e1003044 (2012)).

These studies must be cited and discussed. Specifically, it is important to link the model presented to these studies, as they provide a potential assay to validate the model (in particular Flachs, P. et al. PLoS Genet 8, e1003044 (2012) where both the removal of an allele or increased gene dosage can improve the fertility of B6xSTUS hybrid). Although the in vivo parameters are not known in the hybrids that have been analyzed, the authors may be able to approximate some conditions that are simulated, and the model may propose an interpretation for these observations.

Also, the context of C57BL/6 mice seems a good example of a context where Prdm9 binding becomes limiting as seen in the B6 +/- mice (Baker, C.L. et al. Multimer Formation Explains Allelic Suppression of PRDM9 Recombination Hotspots. PLoS Genet 11, e1005512 (2015)). This seems to indicate that B6 sites have been strongly eroded.

In the discussion, the authors refer to the number of DSBs per meiosis (L701). If DSBs are indeed lower in B6 compared to PWD, it would suggest that the impact is global rather than specific to symmetric/asymmetric site balance (unless an additional hypothesis of co-evolution is invoked).

In addition to the reduction of DSB activity (at symmetric sites and maybe also at asymmetric sites), it has been shown in this context and other contexts (hybrids) that default sites are used. Even if not integrated into the model, this aspect should be discussed. At some point, during Prdm9 evolution, default sites are able to take over, thus this is a very important feature that needs to be taken into consideration/discussed, even if it does not directly impact the present study and model.

*Reviewer #2 (Recommendations for the authors):*

1) The introduction is a bit long but it is maybe necessary as the topic is complex. The authors did a very good job to explain such complexity but maybe so parts could be a bit reduced. For example, between l55-62 could be removed as it is not really discussed later on.

2) As explained above there are several points that should be clarified to help the reader

– In the two first results parts there is no direct result about the evolutionary dynamics per se but some logical predictions based on the fitness landscapes, similar to an invasion analysis (for example, under which conditions a new allele can invade? are there fixed points and which ones?). This step is fully justified but the objective should be explained clearly.

– In figures 2 and 3 it would be good to explain from which equations the figures come. For example, in figure 2 it is not clear whether it corresponds to fitness and probability of binding at an individual or population scale and what is the composition of the population.

– In figure 3 the composition of the population is not clear either: one resident PRDM9 allele? Several? So, "homozygote vs heterozygote genotypes" is not clear (as in Figure 3). We globally understand the rationale but then we need to guess to reconstruct the puzzle from the different pieces of equations.

– l. 310. "These last two assumptions are key, as can be understood by considering how fitness would change over time without them, under the simplest possible scenario, in which all individuals are homozygous for a single PRDM9 allele and all binding sites have the same binding affinity." This statement is rather implicit and the reader may think that the answer is in the sentence (however one must read the next paragraph to fully understand). Maybe rephrase a more explicit explanation.

– In the third part, mutations on PRDM9 are introduced but not before. This should be explained in the model part that the mutation dynamics of PRDM9 are not considered first for the fitness analysis. Otherwise, it leaves the reader with unanswered questions about it until l426. We also only understand when reading this part that the diversity of PRDM9 will be followed. So, this should be explained earlier also.

3) Choice of parameters

– The number of hotspots associated with a new PRDM9 allele is drawn in a uniform distribution between 1 and 5000. This is a bit surprising. Is there any justification for this choice? Instead, we might have expected a unimodal distribution centered for example on the average number of sequences we can find at random in a genome for a motif. A few additional simulations with another distribution could be helpful to check whether the results are not dependent too much on this assumption.

– It is also justified to explore only the case of two heats and the assumptions of many weak sites seem reasonable, but it is not justified. Is the underlying justification that for a given motif of k nucleotides there is 3^k possible one-step mutant motif (so many more) that should only bind weakly (because of one mutant site)? – Among other choices of parameters, the authors decided to consider only two heats (and discussed this choice). However, we can consider that there are only two heats per PRDM9 alleles but that they have different heats, especially the hottest one. If possible, a complementary set of simulations would be to fix n2 and k2 as the authors did but then instead of fixing k1 and considering that each new PRDM9 allele has a different n1, doing the reverse: fixing n1 and drawing k1 in a distribution.

4) Discussion

A possible suggestion would be to summarize observations to explain in a table in parallel with predictions of this model and those of previous red-queen models to clearly show what is better explained, what is not that different, and what could be tested further.

---

## [Author Response]

Essential revisions:1) Introduction could be reduced.

We have shortened the introduction along the lines suggested by reviewers.

2) The objective of each step of the model is not clearly explained and it is left to the reader to understand where the authors want to go. At first read, it is not clear whether the authors present an analysis of the model or simulation results and why they do that. So, the results part deserves rewriting and re-organization to guide the reader.

We edited the Results section to make the progression of the argument clearer (as detailed below).

3) The choice of key parameters is justified except for Prdm9 occupancy, and potential competition between different Prdm9 variants for DSB activity.

We have now added a justification of these choices. We also added two appendixes that consider alternative assumptions, and parts that address specific assumptions, including those about competition, to the Discussion. We now clarify in the Model that our assumption about the vast majority of expressed PRDM9 molecules being bound results in negligible degrees of competition between different PRDM9 variants for DSB activity:

“This approximation assumes that conditional on being bound, PRDM9 molecules arising from different alleles are equally likely to recruit the DSB machinery, but allows for some degree of dominance as a consequence of differences in the numbers of sites bound by each allelic variant genome-wide. However, in the main case we explore below (the "two-heat model"), our assumption that the vast majority of PRDM9 molecules are bound for all alleles results in negligible degrees of dominance arising from differences in genome-wide levels of binding.”

4) Developing the specific predictions relative to previous red-queen models, for Prdm9 diversity in human and mice, for gene dosage effects reported in mice.

We chose the two-heat model as a reasonable first approximation to the true distribution. If we were to consider a more realistic binding distribution (or similarly, if we relaxed our assumption about most PRDM9 molecules being bound), the quantitative conclusions would likely be affected. Accordingly, while our simplified model provides robust insights into the dynamics of PRDM9 evolution, quantities such as the predicted levels of diversity in our model may be off and cannot be readily compared to what is observed in human and mice populations. We now better clarify the scope of our results and what may be done to extend it, in the Discussion.

Re. the second point, we now include a new section in the Discussion, entitled “PRDM9-mediated hybrid sterility” which discusses the reported gene dosage effects in mice.

5) Some more explanation about the difference between the Penrose stair and the Red-queen.

We have expanded our discussion of the Penrose stair model relative to the Red Queen model at the end of the Discussion section “Does the decay of hotspots by GC lead to more or fewer hotspots?” We now make clear that the Penrose stair model is not meant to replace the Red Queen model, but rather, suggests that which direction the Red Queen is running in – towards fewer or more hotspots – is a matter of perspective.

6) Several clarifications on the data presented in the figures are also required.

We clarified all the requested points and added a few others (as detailed below).

Reviewer #1 (Recommendations for the authors):A few specific points also require clarification:1) In the model, the question of competition is not entirely clear.There may be different levels of competition:– Competition for binding of Prdm9 to its sites (this will depend on affinity and protein level);Figure 1A indicates that Prdm9 binds its target as a consequence of competition between sites. It is not very clear what this means. Prdm9 will bind and occupy high-affinity sites more frequently than low-affinity sites. This is not a competition. There may be a competition but only if Prdm9 protein is limiting.

We have changed the text of Figure 1A to read “PRDM9 binds its target sites at rates determined by Michaelis-Menton kinetics” instead of “as a consequence of competition between sites”. In this context, competition refers to the fact that the rate at which PRDM9 will bind and occupy high-affinity sites depends on the number of other sites and their particular affinities (i.e., that sites act as competitive inhibitors of one another). We now explicitly state our assumption that PRDM9 is limiting in the model overview.

The authors write L215. "the corresponding PRDM9 binding sites will always be more likely to be bound in individuals homozygous for the cognate PRDM9 allele relative to heterozygotes": not if Prdm9 protein is not limiting.In fact, by choosing conditions (L354) where 99% of Prdm9 is bound to weak sites in the absence of hotspots, it seems that the authors define conditions where Prdm9 is limiting. What is the rationale for this?

In the context of Equation 1, for any finite and non-zero values of PF and ki, increasing the value of PF will increase the value of Hi (the probability of a site being bound). We have added a paragraph at the end of the model section “Molecular dynamics of PRDM9 binding,” which clarifies that this increase will be small if PRDM9 is non-limiting: “The difference in the probability of binding the cognate PRDM9 in homozygotes and heterozygotes and the degree of competition among binding sites will be small when PRDM9 is not limiting (i.e., when PF≫PT−PF), but can be substantial when PRDM9 is limiting (i.e., when PT≫PF).”

The primary reason for assuming that PRDM9 is limiting is because the gene dosage effects of PRDM9 observed in mice would not occur if PRDM9 expression were not limiting. We now make this point explicitly in the paragraph mentioned above: “Empirical observations suggest that PRDM9 is limiting, with given binding sites bound at substantially greater rates in homozygote individuals than in heterozygotes or hemizygotes (Flachs et al. 2012, Baker et al. 2015)”.

We further assumed that PF≪PT in order to simplify our analysis; as described in Appendix 3, so long as PF≪PT, we can reduce the number of parameters in our model by collapsing k1, k2 and n2 into a single parameter. We clarify this point in the Model section:“In the main case we study below (the “two-heat model”), we assume model parameter values for which the bulk of PRDM9 molecules are bound (i.e., PT≫PF) and thus the competition among binding sites is strongest; this choice simplifies our analysis by reducing the number of parameters that affect the dynamics (see below).”

– Competition between Prdm9 bound and DSB activity (see Diagouraga et al., 2018) (if DSB activity is fixed, then the number of Prdm9 bound may exceed the number of potential DSB, ie 300). This could result in competition between Prdm9 alleles in heterozygotes.

This possibility was briefly mentioned in the Model section “Molecular dynamics of DSBs and symmetric DSBs”. We now underscore the fact that our assumption about most PRDM9 molecules being bound in the absence of hotspots results in negligible degrees of competition between PRDM9 variants for DSB activity there:

“This approximation assumes that conditional on being bound, PRDM9 molecules arising from different alleles are equally likely to recruit the DSB machinery, but allows for some degree of dominance as a consequence of differences in the numbers of sites bound by each allelic variant genome-wide. However, in the main case we explore below (the "two-heat model"), our assumption that the vast majority of PRDM9 molecules are bound for all alleles results in negligible degrees of dominance arising from differences in genome-wide levels of binding.”

2) Figure 1 represents k_on_ and Koff; the text refers to Ki, thus, even if obvious, it should be mentioned that ki=koff/kon.

We now mention it in the text (“Molecular dynamics of PRDM9 binding”):

“where ki (≡koffkon) is the site’s dissociation constant …”

3) Figure 3B: the area shaded in red should correspond for the context where the homozygous have lower fitness that the maximal fitness of the heterozygous. Thus, what is the red-shaded area for > 4800 hotspots?

Individuals homozygous for a PRDM9 allele with n1 hotspots can have reduced fitness relative to individual heterozygous for that allele and one with the optimal number of hotspots for heterozygotes when either: (a) n1 is very small, such that an appreciable number of DSBs localize to weak binding sites, or (b) n1 is very large, such that the degree of symmetry at any individual hotspot is reduced. The red-shaded area to the left corresponds to the former, and the red-shaded area to the right the latter. We now clarify this point in the figure legend: “In both panels, the region shaded in light red on the left reflects the case wherein the homozygous individual will have reduced fitness as a consequence of PRDM9 binding to weak binding sites. In panel B, the region on the right reflects the case wherein the homozygous individual will have reduced fitness as a consequence of limited symmetric binding at hotspots.”

4) Figure 5: The label of the y-axis is missing, please clarify. It would help to describe the figure in the text (L480-484) as way more consistent with the figure itself, and to better explain how the data from Figure 4 was used to derive these hotspot distributions.

We have added a y-axis label to Figure 5, which reads “Probability density”

I assume the initial and final hotspots correspond to exiting and invading alleles? Is this correct?

Yes. We now clarify this point in the Figure 5 figure caption, “The relative densities of initial and final numbers of hotspots (corresponding to invading and exiting alleles, respectively) …” and in the main text when introducing Figure 5 (see below).

"…the distribution is centered close to the optimal number of hotspots.", I guess the authors refer to the invading alleles distribution, please clarify in the text.

We now clarify this point in the text: “The distribution primarily reflects the number of hotspots of successfully invading alleles, and its mean is therefore near the optimal number of hotspots for heterozygotes…”

5) Estimation of the number of Prdm9 bound sites.The authors use the value that has been published by Baker et al. Genome Res 24, 724-32 (2014). However, the problem is that this number is questionable.Baker et al. indicated:"…we estimated that there are;4700 +/- 400 PRDM9-modified sites in the B6 background in an average meiosis." "For estimating the total number of PRDM9-dependent hotspots per meiosis, an equal volume of MNase-digested input DNA was analyzed alongside the ChIP DNA".This method is not correct: by using this Mnase-digested input DNA, the amount of DNA is likely underestimated, as most input DNA molecules are not bound by Prdm9, do not have positioned nucleosomes at Prdm9 binding sites, and thus not protected from Mnase (depending on the duration of treatment) as positioned nucleosomes are. This unknown underestimation factor means that the number of sites bound by Prdm9 is overestimated.The way to address this issue is to design models with different values for pT (from 5000 to 500 for instance). Does it affect the simulations? How?

We thank the reviewer for the suggestion. We have run additional simulations (when considering strong and weak hotspots, k1 = 5 or 50, and when considering large and small population sizes, N=106or103), using PT = 500, 1000 and 2500. The results of these simulations are included and discussed in Appendix 4. We show in Appendix 4 Figure 1 that changing the value of PT results in minimal changes to the qualitative dynamics of our model. The quantitative changes to the model are shown in Appendix 4 Figure 2.

6) Connections with in vivo data should be made:Several important studies in the analysis of Prdm9 hybrid incompatibility have reported dosage effects (ie Mihola et al. Science 323, 373-5 (2009); Flachs, P. et al. PLoS Genet 8, e1003044 (2012)).These studies must be cited and discussed. Specifically, it is important to link the model presented to these studies, as they provide a potential assay to validate the model (in particular Flachs, P. et al. PLoS Genet 8, e1003044 (2012) where both the removal of an allele or increased gene dosage can improve the fertility of B6xSTUS hybrid). Although the in vivo parameters are not known in the hybrids that have been analyzed, the authors may be able to approximate some conditions that are simulated, and the model may propose an interpretation for these observations.

We are grateful for the suggestion, as we believe that this study is a nice example of where our model can explain results in a manner that previous models could not. Specifically, we now include an explanation of the observed gene dosage effects in mice in the Discussion section: “PRDM9-mediated hybrid sterility”.

Also, the context of C57BL/6 mice seems a good example of a context where Prdm9 binding becomes limiting as seen in the B6 +/- mice (Baker, C.L. et al. Multimer Formation Explains Allelic Suppression of PRDM9 Recombination Hotspots. PLoS Genet 11, e1005512 (2015)). This seems to indicate that B6 sites have been strongly eroded.

We now include this reference in support of the notion that PRDM9 is limiting:

“Empirical observations suggest that PRDM9 is limiting, with given binding sites bound at substantially greater rates in homozygote individuals than in heterozygotes or hemizygotes (Flachs et al. 2012, Baker et al. 2015)”.

In the discussion, the authors refer to the number of DSBs per meiosis (L701). If DSBs are indeed lower in B6 compared to PWD, it would suggest that the impact is global rather than specific to symmetric/asymmetric site balance (unless an additional hypothesis of co-evolution is invoked).In addition to the reduction of DSB activity (at symmetric sites and maybe also at asymmetric sites), it has been shown in this context and other contexts (hybrids) that default sites are used. Even if not integrated into the model, this aspect should be discussed. At some point, during Prdm9 evolution, default sites are able to take over, thus this is a very important feature that needs to be taken into consideration/discussed, even if it does not directly impact the present study and model.

Thank you for your suggestions. We now explicitly mention in the Discussion section “Why use PRDM9?” that the use of default sites is also seen in hybrid mice: “In vertebrate lineages in which PRDM9 has been lost, such as canids, birds or percomorph fish, as well as in PRDM9 knockout mice and rats, and to a much lesser degree in sterile mice hybrids, recombination events are concentrated in hotspots that tend to be found associated with promoter-like features…”

In that section, we have expanded our discussion regarding why PRDM9 may have been lost in favor of the use of ‘default’ hotspots.

Reviewer #2 (Recommendations for the authors):1) The introduction is a bit long but it is maybe necessary as the topic is complex. The authors did a very good job to explain such complexity but maybe so parts could be a bit reduced. For example, between l55-62 could be removed as it is not really discussed later on.

Following this suggestion and that of reviewer 1, we have shortened the introduction.

2) As explained above there are several points that should be clarified to help the reader– In the two first results parts there is no direct result about the evolutionary dynamics per se but some logical predictions based on the fitness landscapes, similar to an invasion analysis (for example, under which conditions a new allele can invade? are there fixed points and which ones?). This step is fully justified but the objective should be explained clearly.

We edited the Results section to make the progression of the argument clearer.

– In figures 2 and 3 it would be good to explain from which equations the figures come. For example, in figure 2 it is not clear whether it corresponds to fitness and probability of binding at an individual or population scale and what is the composition of the population.– In figure 3 the composition of the population is not clear either: one resident PRDM9 allele? Several? So, "homozygote vs heterozygote genotypes" is not clear (as in Figure 3). We globally understand the rationale but then we need to guess to reconstruct the puzzle from the different pieces of equations.

We now clarify in the Figure 2 caption that we are “assuming that all individuals are homozygous for a single PRDM9 allele (without turnover)” and thus that the value is the same on the individual or population scale. In the caption of Figure 3, we now clarify that we are showing results “for an individual homozygous for a PRDM9 allele (blue), or heterozygous for two PRDM9 alleles with the same number of hotspots (red)…”

– l. 310. "These last two assumptions are key, as can be understood by considering how fitness would change over time without them, under the simplest possible scenario, in which all individuals are homozygous for a single PRDM9 allele and all binding sites have the same binding affinity." This statement is rather implicit and the reader may think that the answer is in the sentence (however one must read the next paragraph to fully understand). Maybe rephrase a more explicit explanation.

We have re-written this sentence to signpost that we are providing an answer later in the text. It now reads: “In order to illustrate the importance of these assumptions, we first consider how fitness would change over time with or without these assumptions, under the simplest possible scenario, in which all individuals are homozygous for a single PRDM9 allele and all binding sites have the same binding affinity”

– In the third part, mutations on PRDM9 are introduced but not before. This should be explained in the model part that the mutation dynamics of PRDM9 are not considered first for the fitness analysis. Otherwise, it leaves the reader with unanswered questions about it until l426. We also only understand when reading this part that the diversity of PRDM9 will be followed. So, this should be explained earlier also.

We now make it explicit in the second Results section that we are not considering the evolutionary dynamics there, but that they are presented below: “At this point, we do not yet consider the evolutionary dynamics, but gain insights helpful to elucidating these dynamics below.”

3) Choice of parameters– The number of hotspots associated with a new PRDM9 allele is drawn in a uniform distribution between 1 and 5000. This is a bit surprising. Is there any justification for this choice? Instead, we might have expected a unimodal distribution centered for example on the average number of sequences we can find at random in a genome for a motif. A few additional simulations with another distribution could be helpful to check whether the results are not dependent too much on this assumption.

Thank you for your suggestion. We have now added an additional appendix (Appendix 5), which investigates the dynamics of our model when newly arising PRDM9 alleles are initiated with hotspot numbers set near values that would be reasonable for perfect matches to motifs with 10 or 11 non-degenerate bases. We show that this sometimes affects the dynamics (compared to the case in the main text), but when it does, the differences can be readily understood using the same kind of reasoning developed in the main text.

– It is also justified to explore only the case of two heats and the assumptions of many weak sites seem reasonable, but it is not justified. Is the underlying justification that for a given motif of k nucleotides there is 3^k possible one-step mutant motif (so many more) that should only bind weakly (because of one mutant site)?

When the two-heat model is first introduced, it is motivated by being more realistic than the single-heat model (see last paragraph of the section, “Fitness in models with one heat”). The qualifications of its reality are introduced in several following steps. First, in the section "Dynamics of the two-heat model", we point to the two extensions explored in the new Appendices, including one that is based on the model that the reviewer alludes to. Then, in the discussion, we explicitly consider the assumption of two heats compared to a more realistic distribution, as well as the assumption about most PRDM9 molecules being bound. We think that this gradual elaboration is better than trying to 'over-justify' the two-heat model off the bat.

– Among other choices of parameters, the authors decided to consider only two heats (and discussed this choice). However, we can consider that there are only two heats per PRDM9 alleles but that they have different heats, especially the hottest one. If possible, a complementary set of simulations would be to fix n2 and k2 as the authors did but then instead of fixing k1 and considering that each new PRDM9 allele has a different n1, doing the reverse: fixing n1 and drawing k1 in a distribution.

We agree with the reviewer that there are many more variations of the model that could be considered. To that end, we also provide documented code for all our results, making it as easy as possible for other investigators to explore variations on our assumptions. We chose to limit our scope to what we see as the simplest kind of model that captures many of the essential parts that affect PRDM9 evolution.

With regards to the reviewer’s specific suggestion, given two PRDM9 alleles with the same number of hotspots, the one whose hotspots are hotter will always have a higher marginal/allelic fitness, because a greater proportion of PRDM9 will be found bound to hotspots, as opposed to weak binding sites. Accordingly, selection will always favor lower values of k1. This behavior is illustrated in Figure 6D.

4) DiscussionA possible suggestion would be to summarize observations to explain in a table in parallel with predictions of this model and those of previous red-queen models to clearly show what is better explained, what is not that different, and what could be tested further.

While our simplified model provides robust insights into the dynamics of PRDM9 evolution, quantities such as the predicted levels of diversity in our model may be off and cannot be readily compared to what is observed in human and mice populations. We now better clarify the scope of our results and what may be done to extend it, in the Discussion.